



# Low-level jets in the southern North Sea: implications for wind turbine performance using Doppler lidar observations

Pauline Haezebrouck[1], Elsa Dieudonné[1], Anton Sokolov[1], Hervé Delbarre[1], Patrick Augustin[1], and Marc Fourmentin[1]

[1]Laboratoire de Physico-Chimie de l'Atmosphère (LPCA), Université du Littoral Côte d'Opale (ULCO), Dunkerque, France.

**Correspondence:** Pauline Haezebrouck (pauline.haezebrouck@univ-littoral.fr)

**Abstract.**

Accurate knowledge of wind conditions experienced by wind turbines is essential to assess their performance. Among these conditions, low-level jets (LLJs) are important to consider since they have a direct impact on wind turbines, as their cores are frequently located within the rotor layer. In this context, this study investigates the characteristics of LLJs and their formation
mechanisms using 3.3 years of long-range Doppler lidar measurements obtained at Dunkerque, a coastal city on the southern North Sea. In addition, data from an ultrasonic anemometer and from the ERA5 reanalyses of the European Centre for Medium-Range Weather Forecasts (ECMWF) were used to determine the conditions favoring LLJ occurrence. The analysis revealed that LLJs were present in 15.6 % of the 117,411 measured wind profiles. The average jet core speed was 8.4 m s$^{-1}$, with a mean core height of 267 m. LLJs were more frequent during nighttime, especially in spring and summer. These characteristics
were consistent with those obtained at other sites in the North Sea region, with some differences attributable to the location of Dunkerque on the coast near the Dover Strait. This position introduced additional formation mechanisms for LLJs, including land–sea thermal gradients and wind channeling in the English Channel. The impact of LLJs on wind turbines of varying dimensions was then assessed for both energy production and structural loads. For conventional turbines, with a hub height around 100 m, LLJs counter-intuitively tend to decrease power production at high wind speeds. Conversely, for more recent
and future wind turbines, LLJs will improve power production in all conditions. The increase in turbine size will also greatly reduce their exposure to detrimental wind shear conditions, both in terms of speed and direction shear.

## 1 Introduction

Low-level jets (LLJs) are wind maxima in the lower layers of the troposphere, typically occurring at heights ranging between 80 and 1,500 m (Blackadar, 1957). This meteorological phenomenon is known to impact multiple sectors, notably influencing
regional weather, aviation safety, air quality and wildfires (Wu and Raman, 1998; Golding, 2005; Sharples, 2009; Klein et al., 2019). Recently, the frequency of LLJs has become a critical factor to consider when selecting wind energy production sites, as LLJs complicate both forecasting and energy production due to the fluctuations in wind speed and wind direction that they cause (Nunalee and Basu, 2014). Technological advancements allow the design of higher and more efficient wind turbines, especially offshore (Mehta et al., 2024). As a consequence, turbine rotors are increasingly interacting with LLJs (Pichugina



et al., 2017) and the associated wind shear, i.e., the vertical gradient in wind speed and wind direction that LLJs induce
(Gutierrez et al., 2014; Barthelmie et al., 2020). Besides impacting the energy harvesting of isolated rotors (Weide Luiz and
Fiedler, 2022), changes in the wind profile shape also influence wake recovery and, therefore, the overall wind farm production
(Na et al., 2018; Doosttalab et al., 2020; Gadde and Stevens, 2021). Moreover, jet wind shear contributes to turbine blade
fatigue by imposing mechanical stress and repetitive loads, leading to structural wear over time (Gutierrez et al., 2017, 2019).
Therefore, evaluating LLJs' characteristics and determining the mechanisms driving their formation is essential to correctly
assess loads and power production.

The North Sea is a region of particular concern regarding the influence of LLJs on wind turbines, as multiple wind farms
are already operational, and the potential for further development remains important. Several studies have focused on charac-
terizing LLJs in this area, aiming to better understand the frequency, intensity, and height of LLJs, as well as the mechanisms
behind their formation. Measurements from offshore platforms and along the German, Dutch and French coasts indicate that
LLJs occur 3.8 % to 14.5 % of the time in the North Sea (Baas et al., 2009; Kalverla et al., 2017; Duncan, 2018; Wagner et al.,
2019; Rausch et al., 2022; Dieudonné et al., 2023). The jet detection height in these studies was limited to an altitude of 300
to 500 m, most frequently because they relied on short-range wind profiles. However, jet detection and the resulting character-
ization of jet property distributions are highly dependent on the vertical measurement range of the instruments (Kalverla et al.,
40    2019).

Aside from the jet characteristics, several studies investigated the jet formation mechanisms existing in the North Sea (Baas
et al., 2009; Kalverla et al., 2017; Wagner et al., 2019; Dieudonné et al., 2023). Frictional decoupling is one of these mech-
anisms; it typically occurs at night, when radiative cooling leads to a stable boundary layer, reducing the impact of surface
friction on higher-level winds (Blackadar, 1957; Bonner, 1968; Davis, 2000; Baas et al., 2009; Van de Wiel et al., 2010).
Frictional decoupling can also occur when a warm air flow is advected over much colder water (Smedman et al., 1993). In
coastal areas, the daily cycle of the land-sea temperature gradient also induces LLJs, namely sea and land breezes (Chao, 1985;
Kottmeier et al., 2000; Soares et al., 2014). Moreover, the specific position of Dunkerque at the entrance of the English Channel
induces an additional formation mechanism associated with wind channeling in the Dover Strait (Capon, 2003).

The upcoming offshore wind farm deployment near Dunkerque (https://parc-eolien-en-mer-de-dunkerque.fr/project-en/) has
driven the need to investigate LLJs in this coastal city on the southern North Sea. While a first study based on 4.3 years of
short-range lidar data provided insights into LLJ characteristics up to 300 m in the region (Dieudonné et al., 2023), the present
work extends this analysis to higher altitudes using 3.3 years of Doppler lidar observations up to 1,500 m. This extended range
allows for a complete characterization of LLJs in Dunkerque and a comprehensive assessment of their impacts on wind turbines
regarding power production and structural loading, since jets can affect wind turbine performance even when located above the
rotor layer. Three turbine sizes were investigated to quantify how LLJs' impacts evolve with the increasingly larger structures
being deployed in modern wind farms.

The paper is structured as follows: Sect. 2 describes the study site, the instrumentation, the data processing and the jet de-
tection methodology, in consideration of the higher vertical extent of the instruments. The results concerning the jet properties,





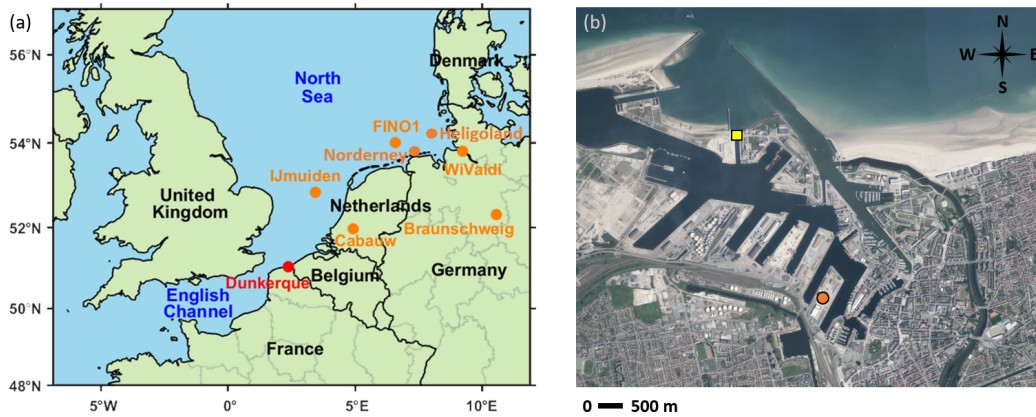

**Figure 1.** (a) Location of Dunkerque on the southern part of the North Sea, and locations of previous studies on low-level jets in the region (orange dots). (b) Position of the lidars (orange dot) and the ultrasonic anemometer (yellow square)(background image from https://www.geoportail.gouv.fr/).

their formation mechanisms and their impact on wind turbines are presented in Sect. 3 and discussed in Sect. 4. A summary of
the work and a conclusion on the results are provided in Sect. 5.

## 2 Methodology

### 2.1 Observation site, experimental setup and data availability

The study was performed in Dunkerque (51.04° N, 2.37° E), a coastal city in northern France, bordered by the North Sea and located at the entrance to the English Channel (Fig. 1a). The coastline in this region is oriented south-southwest to east-
northeast. The surrounding terrain is predominantly flat, with an average elevation of 5 m above mean sea level (AMSL).

The wind profiles up to 1,500 m were obtained by using successively two long-range Doppler lidars (Scanning Wind Cube WCS100 and WCS200, from Vaisala, France). These instruments are nearly identical and differ only in their laser power, which does not affect the measurements since the vertical extent of the observations is actually limited by the presence of aerosols. Both lidars were installed on top of a building at 14 m AMSL and located approximately 1.6 km aback from the
coastline (Fig. 1b). In this work, the wind profiles were reconstructed using vertical scans of the Range-Height-Indicator (RHI) type, differing from the commonly used Doppler Beam Swinging (DBS) method. Pairs of perpendicular RHI sweeps in the cardinal directions (Table 1) were used to retrieve the profiles of the meridional and zonal components of the wind. During the reconstruction process, the vertical wind was assumed to be negligible when converting the radial wind into horizontal wind. The resulting wind components were then averaged over horizontal layers and combined to obtain the horizontal wind speed
and direction in each layer, giving the complete wind profile (Banta et al., 2002; Bonin et al., 2017; Dieudonné et al., 2025). This technique enables the observation of low-altitude jets by eliminating the blind zone near the ground, which extends up



**Table 1.** Characteristics of the Range-Height-Indicator (RHI) scans.

| | |
|---|---|
| Azimuth | $0°$ and $90°$ |
| Elevation range | $0°$ to $180°$ with a $2°$ resolution |
| Axial range | 50 m to 3.3 km with a 25 m resolution |
| Duration | 90 s per sweep, 3 min 10 s for the pair |
| Repetition period | 15 minutes |

to 50 m above the instrument height with DBS scans. Moreover, for a similar observation time, the resulting profiles are less noisy, which is beneficial for a more accurate jet detection.

Prior to reconstructing the horizontal wind profile from the RHI sweeps, data points were filtered by applying a Carrier-to-Noise Ratio (CNR) threshold, requiring a minimum value of $-27$ dB. Additionally, radial wind speeds outside of the measurement scope of the instruments ($\pm\,30\,\mathrm{m\,s^{-1}}$) were deleted. In some cases, very high wind speeds were observed in RHI scans, resulting from range-folded echoes. These spurious values were eliminated by applying a filter comparing each data point to its surrounding values, identifying artifacts and discarding them (Dieudonné et al., 2025).

The measurements cover the period from 2018 to 2024, with data provided by the WCS100 between 2018 and late 2022, as well as for 2024, and by the WC200 in 2023. The total number of measurements consists of 117,411 profiles, corresponding to 1,204 days and 3.29 years of data. Gaps in the dataset were caused by field campaigns on other sites and maintenance activities. The detail of these periods is given in Fig. A1a in the Appendix. Although each month is represented in the dataset, data availability varies across the years, affecting the LLJ statistics, particularly the assessment of the annual variability of LLJs in the region. November and December showed the highest and lowest data availability, with 66 % and 31 % of the time, respectively. To address this issue, the number of LLJ wind profiles was adjusted by dividing the observed values by the corresponding fraction of available data. The details of profile numbers before and after correction are given in Table A1. Data availability also varied with altitude, as the lidar range is reduced in case of rainy or cloudy conditions. At 100 m, the data availability was 97 %, but this value decreased to 86 % at 500 m, 60 % at 1,000 m, and further dropped to 26 % at 1,500 m (Fig. A1b).

A three-dimensional ultrasonic anemometer (Metek USA-1, from Metek GmbH, Germany) was used to link LLJ occurrences to atmospheric conditions, especially stability. It was located in the port, 1.8 km from the lidars' location (Fig. 1b), atop a 15-m mast. The anemometer observations, which cover the whole period from 2018 to late 2024, were recorded at 20 Hz and averaged on 15-minute periods, corresponding to the temporal resolution of the wind profiles. Even though a temporal offset between the lidar and sonic measurements was observed, no interpolation was performed, and the closest measurement was used to correlate the data.





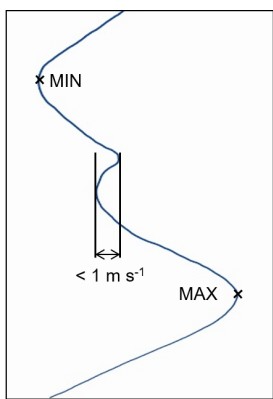

**Figure 2.** Illustration of the criterion for the minimum selection: a minimum is neglected when it is followed by an increase in wind speed of less than $1\,\mathrm{m\,s^{-1}}$ before dropping below the minimum value again.

## 2.2 Identification of low-level jet profiles

In this work, LLJs were defined as wind speed maxima associated with a wind speed fall-off of at least $2\,\mathrm{m\,s^{-1}}$ and 20 % compared to the minimum above. These criteria were selected as they are commonly used in other studies focusing on LLJs in the North Sea (Baas et al., 2009; Kalverla et al., 2017; Duncan, 2018; Wagner et al., 2019; Dieudonné et al., 2023), which enabled
a better comparison of the results. To deal with missing data in the profiles, especially at high altitudes, any measurements above gaps exceeding 100 m were excluded from the jet detection process. Moreover, to avoid fluctuations interfering with jet detection, when a minimum was followed by an increase in wind speed of less than $1\,\mathrm{m\,s^{-1}}$ before dropping again (Fig. 2), this minimum was neglected and a higher-altitude minimum was considered instead (Baas et al., 2009).

To ensure that detected jet events were persistent over time, a continuity criterion was introduced, requiring the jet to last for
a minimum of 1 hour and 30 minutes (Baas et al., 2009), corresponding to 6 consecutive 15-minute profiles. Other studies used shorter durations of 20 minutes (Dieudonné et al., 2023), 30 minutes (Ziemann et al., 2020; Rausch et al., 2022; Weide Luiz and Fiedler, 2022), or 1 hour (Tuononen et al., 2017). In this study, a longer criterion was considered as it improved the jet detection by excluding isolated jets. Additionally, LLJ core heights were required not to vary by more than 50 m between two consecutive profiles. The effect of these different criteria on jet detection is presented in Sect. 3.1.

## 2.3 Weather conditions

In order to relate atmospheric conditions to LLJ occurrence, the atmospheric stability was assessed using the Monin-Obukhov stability parameter $\Lambda$, defined as:

$$\Lambda = -\frac{\kappa\,g\,\overline{w'T'}}{\overline{T}\,u_*^3} \tag{1}$$

With $\kappa$ the von Karman constant, $g$ the gravity acceleration, $\overline{w'T'}$ the turbulent sensible heat flux, $\overline{T}$ the average temperature in kelvin, and $u_*$ the friction velocity. All the parameters were computed using the ultrasonic anemometer. To ensure



**Table 2.** Hub heights and rotor diameters of the wind turbine models considered for assessing the impact of low-level jets on turbine performance.

| Wind turbine | Conventional | Advanced | Future |
|---|---|---|---|
| Hub height (m) | 105 | 160 | 195 |
| Rotor diameter (m) | 164 | 280 | 350 |

result comparability with the previous study conducted in Dunkerque (Dieudonné et al., 2023), the same thresholds for classifying atmospheric stability into modified Pasquill stability classes were used, namely $\Lambda < -0.072$ m$^{-1}$ for extremely unstable, $-0.072 \leq \Lambda < -0.015$ m$^{-1}$ for moderately unstable, $-0.015 \leq \Lambda < -0.002$ m$^{-1}$ for slightly unstable, $-0.002 \leq \Lambda < 0.003$ m$^{-1}$ for neutral, $0.003 \leq \Lambda < 0.016$ m$^{-1}$ for moderately stable, and $\Lambda \geq 0.016$ m$^{-1}$ for extremely stable conditions (Xiang, 2011).

Additionally, weather data from the European Centre for Medium-Range Weather Forecast (ECMWF) reanalyses ERA5 (Hersbach et al., 2020) were used to derive the conditions favoring the formation of LLJs. ERA5 spatial resolution is $0.25°$, and the temporal resolution of 1 hour was interpolated to align with the timing of the measured wind profiles. The land-sea temperature gradient was computed using the 2-m temperature extracted at two points along a transect perpendicular to the coastline, with both points located approximately 23 km from the shoreline ($51.25°$ N, $2.25°$ E offshore and $50.85°$ N, $2.5°$ E onshore). These locations were selected to match those used in the previous study in Dunkerque (Dieudonné et al., 2023), allowing for a comparison of the results.

## 2.4 Jet-turbine interaction

To determine the effect of LLJs on energy production, the entire wind profile within the rotor area was used, as a single measurement point at the turbine hub is not sufficient to accurately determine the power output of a wind turbine (Antoniou et al., 2009). This is particularly relevant for turbines experiencing LLJs, where the error caused by using a single measurement point increases as the rotor size of the turbine grows. Three wind turbine models (Table 2) were selected based on the characteristics of (i) the most common offshore turbines in the North Sea, (ii) the most advanced turbine model currently available, and (iii) the turbine model projected for future uses (Global Wind Energy Council, 2024).

The power production of the different wind turbine models under LLJ and non-LLJ conditions was computed using the following equation:

$$P = \frac{1}{2}\rho A C_p U_{eq}^3 \tag{2}$$

Where $\rho = 1.2$ kg m$^{-3}$ is the air density, $A$ is the rotor-swept area, and $C_p$ is the power coefficient taken as its theoretical maximum value of $16/27$ (Betz, 1920). $U_{eq}$ is the equivalent wind speed obtained by weighting the cubic wind speeds across



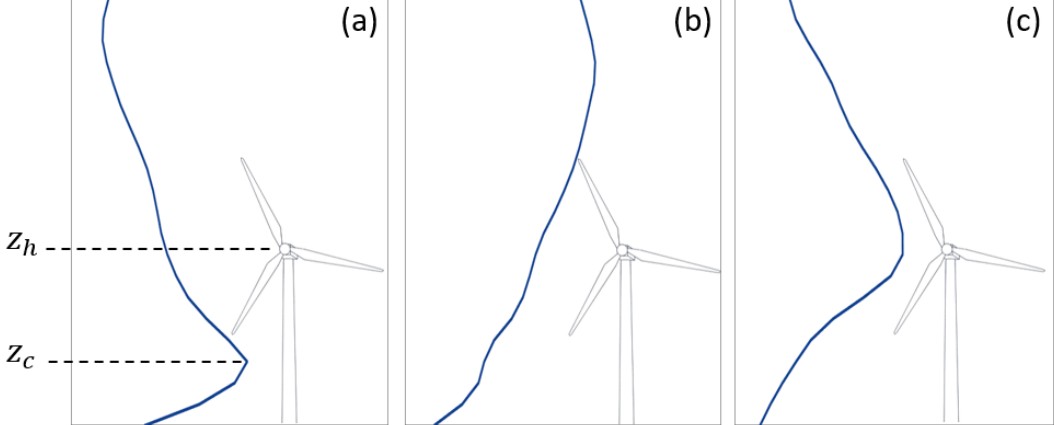

**Figure 3.** Relative position of the low-level jet core with respect to the turbine rotor, resulting in (a) entirely negative shear across the rotor ($\xi \geq 1$), (b) fully positive shear ($\xi \leq -1$), and (c) a combination of both positive and negative shear within the rotor area ($-1 < \xi < 1$).

the entire rotor area (Wagner et al., 2009):

$$U_{eq} = \sqrt[3]{\frac{1}{A}\left(\sum_i \bar{U}_i^3 \cdot A_i\right)} \tag{3}$$

The index $i$ denotes the wind profile measurement heights in the rotor zone, $\bar{U}_i$ are the 15-minute average wind speed values, and $A_i$ are the areas of the corresponding horizontal layers swept by the rotor, while $A$ is again the total rotor-swept area.

LLJs also affect wind turbines due to the shear below and above their cores. This impact varies depending on the sign of the vertical velocity gradient. Negative shear, corresponding to LLJ cores located below the rotor zone (Fig. 3a), has a beneficial effect by reducing wind turbine loads, especially on the nacelle and tower (Gutierrez et al., 2017). On the contrary, positive shear, associated with jet cores situated above the rotor area (Fig. 3b), increases static and dynamic loads of the structure (Gutierrez et al., 2016). In the case of a LLJ core located within the rotor zone, the wind turbine experiences both positive and negative shear (Fig. 3c), leading to a complex interaction between the turbine blades, the tower and the resulting loads (Gutierrez et al., 2019).

To assess the impact of LLJ wind shear on the turbines, an adimensional parameter was introduced by Gutierrez et al. (2017), that represents the LLJ core relative distance to the turbine hub. This parameter $\xi$ is defined as:

$$\xi = \frac{z_h - z_c}{R} \tag{4}$$

Where $z_h$ and $z_c$ are the hub and LLJ core heights, respectively, and $R$ is the rotor radius. Therefore, $\xi \leq -1$ corresponds to entirely positive shear in the rotor zone, $\xi = 0$ is half positive and negative, and $\xi \geq 1$ is completely negative shear.

To further investigate the distribution of shear in the rotor zone, especially considering the increasing size of wind turbine rotors, the shear was divided into two contributions: from the upper blade tip to the hub and from the lower blade tip to the hub.





This prevents misleading zero or low shear values in the case of jets located within the rotor zone, despite significant positive and negative shear applying on the blades. Finally, since directional shear can also impact wind turbines, it was calculated
as the magnitude of the wind direction difference between the highest and lowest points of the turbine rotor, divided by the distance between these positions.

## 3 Results

### 3.1 Validation of the detection method

Different configurations of detection criteria were tested to assess their influence on the fraction of detected jets (Table 3). As
expected, longer time continuity criteria led to slightly fewer identified jets, with the percentage decreasing from 16.8 % for a 30-min criterion to 16.4 % for 1 h, and further to 15.6 % for 1.5 h. The latter value of 1.5 h was selected for the rest of the detection process in order to reduce the fraction of isolated jets observed for shorter durations. With the same objective of excluding isolated jets, a continuity criterion in height between two consecutive profiles was introduced. The detected fraction varied from 20.1 % with no height continuity to 17.7 % when a 100-m threshold was applied and to 10.4 % with a value of
50 m. This latter threshold was retained for the jet detection process.

Initially, a criterion on the wind decrease below the jet was also applied, as considered by some other researchers (Andreas et al., 2000; Karipot et al., 2006; Tuononen et al., 2017). The objective was to reduce the false detection of multiple jets, as the criterion of $1\,\mathrm{m\,s^{-1}}$ for the wind minimum selection may result in two consecutive maxima being associated with the same minimum (Fig. 2). This incorrect association could wrongly lead to the identification of two distinct jets, whereas only a single
jet is physically present. However, this approach reduced the detected jets by 41 %, notably because it resulted in the exclusion of very low-altitude jets, such as sea breezes. As a consequence, no criterion was imposed on the wind decrease below the jet. To address the multiple jet issue, wind maxima associated with the same wind minimum were discarded if the double jet did not persist between two consecutive profiles.

Considering these criteria for the detection, LLJs appear to be non-negligible since they were present on 18,294 wind
profiles over the whole period, corresponding to 15.6 % of the dataset. This percentage is approximately 7 times higher than the proportion of jets detected in the previous study in Dunkerque, when using the same continuity criteria (Dieudonné et al., 2023). However, limiting the wind profiles to a 300 m height, i.e. the same altitude as in Dieudonné et al. (2023), resulted in a similar percentage of detected jets – 2.89 % compared to 2.19 % in the previous study – the difference being compatible with the interannual variability. Therefore, jets occurred above 300 m for approximately 13 % of the time. Cases presenting two
simultaneous jets at different altitudes were very rare, representing only 755 wind profiles, i.e., 4.1 % of the jet profiles and 0.64 % of time.





**Table 3.** Number of LLJ profiles detected for different sets of detection criteria: time and height continuity criteria, and criterion on the wind decrease below the jet.

| Time continuity criterion | Height continuity criterion | Wind decrease below the core | Jet detection height | Number of LLJ profiles | Fraction of LLJ profiles (%) |
|---|---|---|---|---|---|
| 30 min | 50 m | No | 1,500 m | 19,747 | 16.8 |
| 1 h | 50 m | No | 1,500 m | 19,303 | 16.4 |
| 1.5 h | 50 m | No | 1,500 m | 18,294 | 15.6 |
| 1.5 h | N/A | No | 1,500 m | 23,624 | 20.1 |
| 1.5 h | 100 m | No | 1,500 m | 20,730 | 17.7 |
| 1.5 h | 50 m | Yes | 1,500 m | 12,246 | 10.4 |
| 30 min | 50 m | No | 300 m | 4,076 | 3.47 |
| 1.5 h | 50 m | No | 300 m | 3,387 | 2.89 |

## 3.2 Jet characteristics

### 3.2.1 Core direction

The wind rose over the whole dataset period is presented in Fig. 4a, at a height of 275 m AMSL, corresponding to the lidar measurement level closest to the mean altitude of the jet cores. It shows that winds from the south-southwest direction were predominant in the general distribution. In contrast, the distribution of the jet core directions (Fig. 4b) showed distinct contributions, which were categorised into five major directional groups. The dominant LLJ core direction was northeast, ranging from 0° to 70° in direction, and accounting for 25 % of the jet profiles. These jets occurred predominantly during daytime, for 83 % of the cases (Fig. 4c). The second group, which represented 22 % of the occurrences, consisted of LLJs with easterly core directions (70° to 130°), that occurred predominantly at night, for 72 % of the cases (Fig. 4d). Southerly LLJs (130° to 220°), which accounted for 24 % of the cases, were also nocturnal, at 74 %. Another group corresponded to southwesterly directions (220° to 300°); it represented 18 % of the LLJs and comprised a significant part of strong jets. Finally, the last group corresponded to northwesterly directions (300° to 360°), accounting for only 11 % of the LLJs, making it the least represented group with mostly low-velocity and diurnal jets.

### 3.2.2 Core speed

The average LLJ core speed was $8.4 \, \mathrm{m \, s^{-1}}$ and the maximal measured value reached $25.1 \, \mathrm{m \, s^{-1}}$; however, only 0.37 % of the LLJs exceeded a velocity of $20 \, \mathrm{m \, s^{-1}}$. Consequently, the velocity distribution of the jet core (Fig. 5) was limited to this value. The distribution indicates that jet speeds were most frequently observed between 4 and $10 \, \mathrm{m \, s^{-1}}$ in Dunkerque.

When detailing this distribution according to LLJ core directions, northeasterly, easterly and southwesterly groups mainly followed the same distribution. On the contrary, it is clear that northwesterly jets peaked at lower speed values. The southerly LLJs showed a broader distribution, although a similar peak was observed. As for the direction, the LLJ core speed was divided

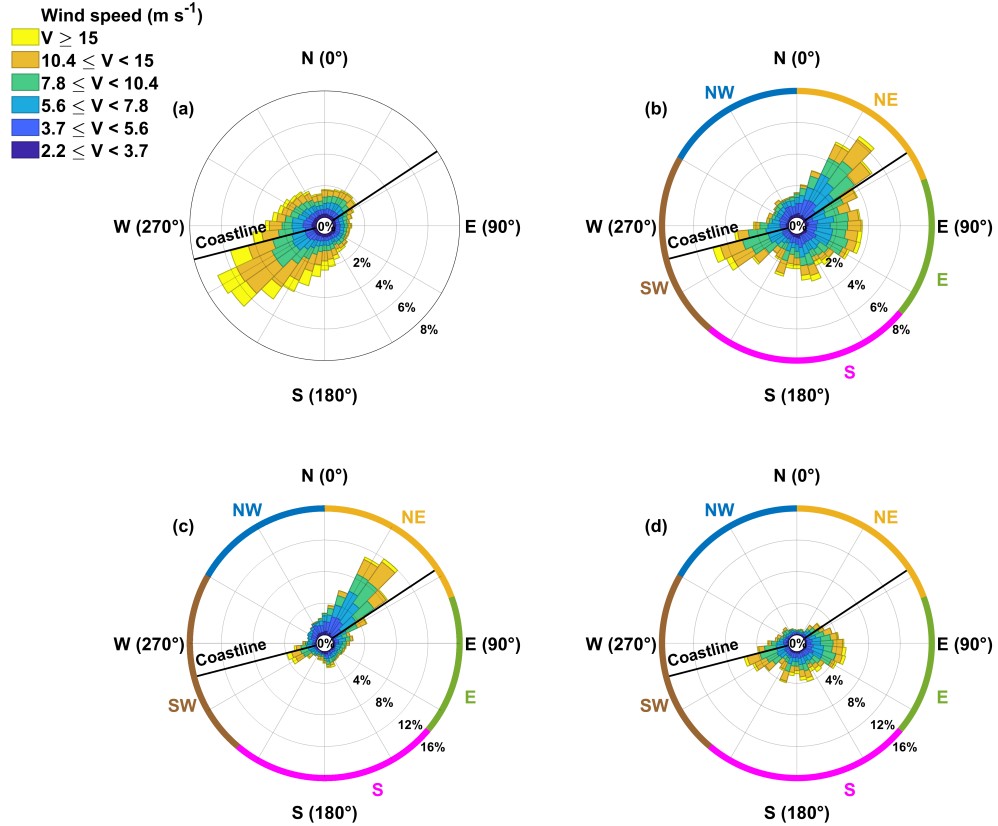

**Figure 4.** (a) General wind rose at 275 m AMSL, (b) wind rose of the low-level jets, (c) wind rose of the daytime low-level jets, and (d) wind rose of the nighttime low-level jets, plotted using the speed groups defined in Sect. 3.2.2. The black line represents the coastline direction, with the sea located on the upper part and the land on the lower part.

into groups. The six wind speed intervals previously used in the wind rose (Fig. 4) are the following: $2.2 \leq V < 3.7 \, \mathrm{m\,s^{-1}}$, $3.7 \leq V < 5.6 \, \mathrm{m\,s^{-1}}$, $5.6 \leq V < 7.8 \, \mathrm{m\,s^{-1}}$, $7.8 \leq V < 10.4 \, \mathrm{m\,s^{-1}}$, $10.4 \leq V < 15.0 \, \mathrm{m\,s^{-1}}$ and $V \geq 15.0 \, \mathrm{m\,s^{-1}}$. These intervals correspond to percentiles of the wind speed distribution: 0–5[th], 5[th]–25[th], 25[th]–50[th], 50[th]–75[th], 75[th]–95[th], and 95[th] and
above.

### 3.2.3 Core altitude

The maximum LLJ core height was measured at 1,375 m, but only 0.28 % of the jet cores reached an altitude above 1,200 m. As a consequence, the core altitude distributions (Fig. 6) were restricted to this value. The LLJs' average altitude was 267 m, but they were most frequently observed at heights ranging from 100 to 200 m. These altitudes correspond to the operational
heights of offshore wind turbines (Wagner et al., 2019). In particular, 44 % of the LLJ cores were located within the rotor-





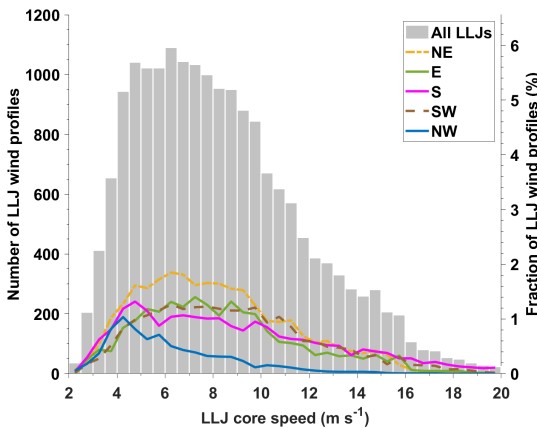

**Figure 5.** Distribution of the low-level jet core speeds detailed by direction classes, as defined in Sect. 3.2.1. Fractions indicated on the right y-axis are calculated as a proportion of the total number of LLJ profiles (18,294).

swept area of conventional wind turbines in the North Sea. This percentage increased for larger wind turbines, reaching 73 % for state-of-the-art turbines and 79 % for future, larger-scale turbines. The resulting impact on wind turbines will be analyzed in Sect. 3.4.

Figure 6a shows the distribution of LLJ core altitudes as a function of their direction. The most frequent core altitudes for all
directions were comprised between 100 m and 200 m, as for the general distribution, except for southwesterly jets occurring more frequently in the 200–300 m range and showing a distribution skewed towards slightly higher altitudes. Northeasterly jets accounted for the largest proportion of very low-altitude jets (below 100 m), representing 39 % of their occurrences. In contrast, easterly and southerly jets were associated with the highest proportion of elevated jets (above 500 m), with 29 % and 26 %, respectively. LLJ core altitude plotted as a function of core speed (Fig. 6b) reveals that low-speed jets were particularly
present at lower altitudes (below 200 m) and were almost absent above this level. On the contrary, strong jets were concentrated at moderate to high altitudes and were absent below 100 m.

### 3.2.4 Annual cycle

The annual cycle of the jet occurrence (Fig. 7) shows that LLJs were predominantly observed from March to October, reaching a peak of 14 % in June. Their frequency diminished considerably during the colder months, from November to February,
with November showing the lowest occurrence of 2 %. Figure 7a presents the annual cycle of LLJs detailed by core direction, showing that northeasterly and easterly LLJs followed the global distribution pattern, with their frequency dropping to near zero between November and February and increasing in the warm months, with the peak of occurrence in June due to northeasterly jets. Northwesterly jets also followed the global distribution; however, they occurred over a shorter period (May to September), similar to the southwesterly jets. In contrast, southerly LLJs were observed throughout the year, with a higher frequency during
the transitions between the cold and warm seasons. The annual distribution of LLJ occurrence detailed by core speed (Fig. 7b)





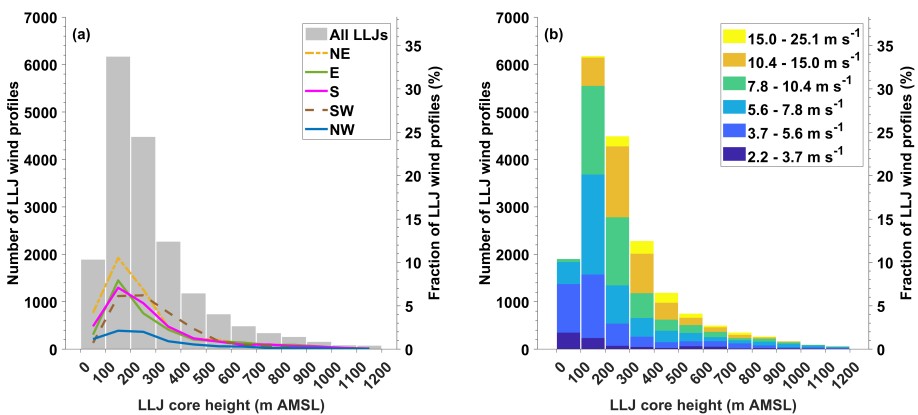

**Figure 6.** Distribution of the low-level jet core heights detailed by (a) core direction, as defined in Sect. 3.2.1, and (b) speed classes, as defined in Sect. 3.2.2. Fractions indicated on the right y-axes are calculated as a proportion of the total number of LLJ profiles (18,294).

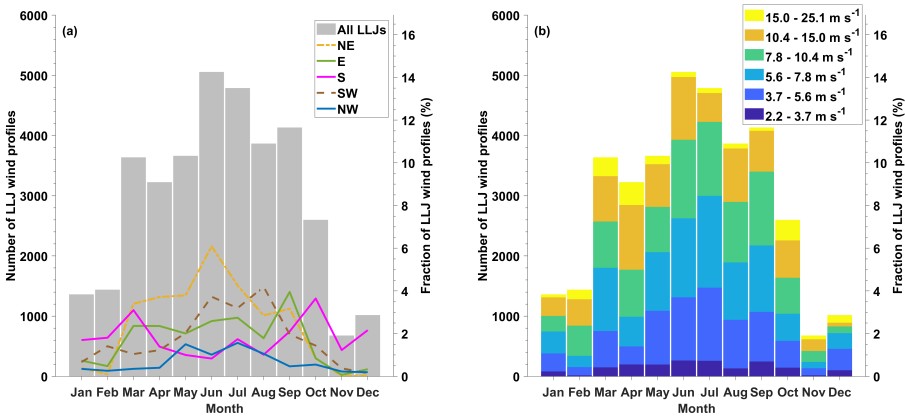

**Figure 7.** Low-level jet annual cycle detailed by (a) core direction, as defined in Sect. 3.2.1, and (b) speed classes, as defined in Sect. 3.2.2. The values are corrected to account for variations in monthly data availability, as described in Sect. 2.1.

indicates that the weakest jets were concentrated during the warm months and generally followed a similar annual distribution as the northeasterly jets. High-speed LLJs occurred year-round, but their peak occurrence was observed in March and October, which corresponds to the periods during which southerly jets were the most frequent.

### 3.2.5 Diurnal cycle

LLJs' occurrence also varied according to a diurnal cycle (Fig. 8), showing that LLJs were more frequent at night. Conversely, the period with the lowest frequency occurred in the morning, between 08:00 and 12:00 UTC, which represented only 9 % of the LLJ profiles. Following this minimum, the jet frequency increased again, reaching a plateau from 12:00 to 16:00 UTC. The



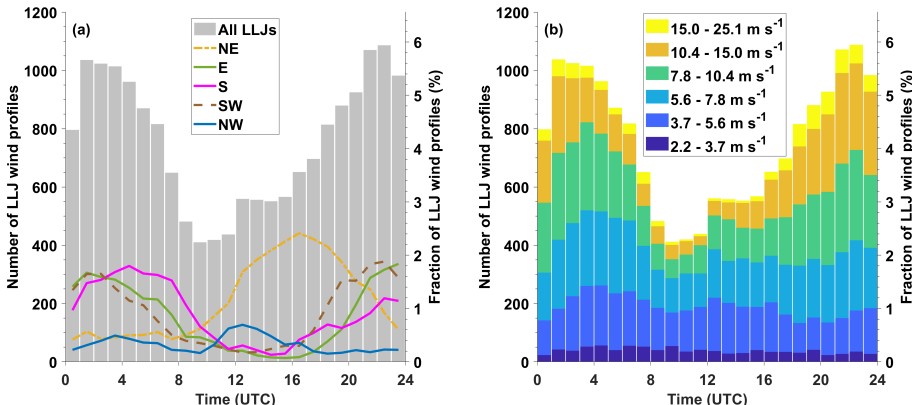

**Figure 8.** Low-level jet diurnal cycle detailed by (a) core direction, as defined in Sect. 3.2.1, and (b) core speed classes, as defined in Sect. 3.2.2. Time is given in the Universal Time Coordinate (UTC) system, which closely corresponds to the local solar time. Fractions indicated on the right y-axes are calculated as a proportion of the total number of LLJ profiles (18,294).

diurnal cycle detailed by jet core directions (Fig. 8a) shows that the afternoon peak mainly corresponded to northeasterly jets. Northwesterly jets also showed a slight increase during this period, with their peak of occurrence in the early afternoon, while

northeasterly LLJs reached their maximum later in the afternoon. Easterly, southerly, and southwesterly LLJs were almost absent during the day, as they predominantly formed in the late afternoon or early evening and remained into the night (17:00 to 08:00 UTC). Southerly jets persisted slightly longer in the morning, dissipating around 11:00 UTC, which follows the fact that they occurred during periods when sunrise happens later in the morning (Sect. 3.2.4). Additionally, a weak contribution of northwesterly jets occurred in the early morning. Figure 8b presents the diurnal cycle of LLJs detailed by core speed.

It indicates that strong LLJs mostly occurred at night, particularly during the first hours after sunset (from 18:00 to 24:00 UTC). In contrast, weak jets were present throughout the day but were slightly more frequent in the early afternoon. A more pronounced peak of weak jets occurred at the end of the night, which may correspond to the gradual weakening of the nocturnal jets before dawn.

### 3.2.6 Event duration

To facilitate comparison with the findings of Dieudonné et al. (2023) on jet duration, events were defined using the same continuity criterion, with LLJs less than 3 hours apart being considered as a single event. As a result, 769 events were identified from the total of 18,294 jet profiles. The longest event lasted 23.6 h, and the average duration was 5.4 h. Figure 9 presents the distribution of LLJ event durations, the first bin covering events lasting from 1.5 h to 2 h, with all subsequent bins having a uniform width of 1 h. The most frequent jets were those lasting between 1.5 and 2 h, accounting for 35 % of all events. The

frequency then decreased with increasing duration, with 37 % of the events lasting between 2 and 5 h, 15 % between 5 and 10 h, and 13 % lasting more than 10 h.





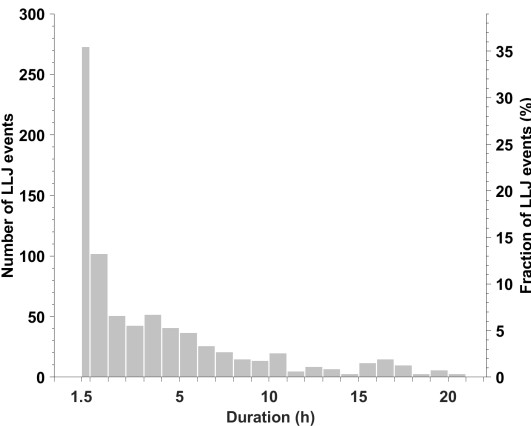

**Figure 9.** Distribution of the low-level jet event durations. Fractions indicated on the right y-axis are calculated as a proportion of the total number of LLJ events (769).

## 3.3 Conditions favoring LLJ formation

### 3.3.1 Atmospheric stability

The general distribution of the Pasquill atmospheric stability classes (Fig. 10a) indicates that the majority of the wind pro-
files fell within the slightly unstable and neutral classes, accounting for 48 % and 31 % of the time, respectively. The same distribution restricted to LLJs profiles (Fig. 10b) shows that the maximum occurrence for all LLJ direction groups coincided with the peak of the general distribution, observed under slightly unstable conditions, and to a lesser extent, neutral conditions. Northeasterly jets represented the highest proportion of jets formed under extremely unstable to moderately unstable conditions. This finding is coherent with the fact that these jets mainly occurred during daytime (Sect. 3.2.5), when the atmospheric
boundary layer is typically unstable. In contrast, easterly and southerly jets were the most frequent under moderately stable to extremely stable conditions, which is consistent with their predominant formation at night. Conversely, northwesterly jets, which occurred over the entire diurnal cycle, followed the general stability distribution and occurred under the full range of conditions.

### 3.3.2 Land-sea temperature gradient

Figure 11 presents the distribution of the temperature difference between land and sea, computed as detailed in Sect. 2.3. Northeasterly jets presented a distinct distribution compared to other directions, showing a clear shift toward positive gradient values. This indicates that such jets predominantly formed when the land was warmer than the sea, which is consistent with their peak of occurrence during the afternoons of the summer months. A similar pattern was in part observed for northwesterly jets. However, these jets also developed when sea temperature exceeded land temperature, for gradient values ranging between -4 K
and 0 K, suggesting their formation involved different mechanisms. On the contrary, easterly, southerly and southwesterly LLJs





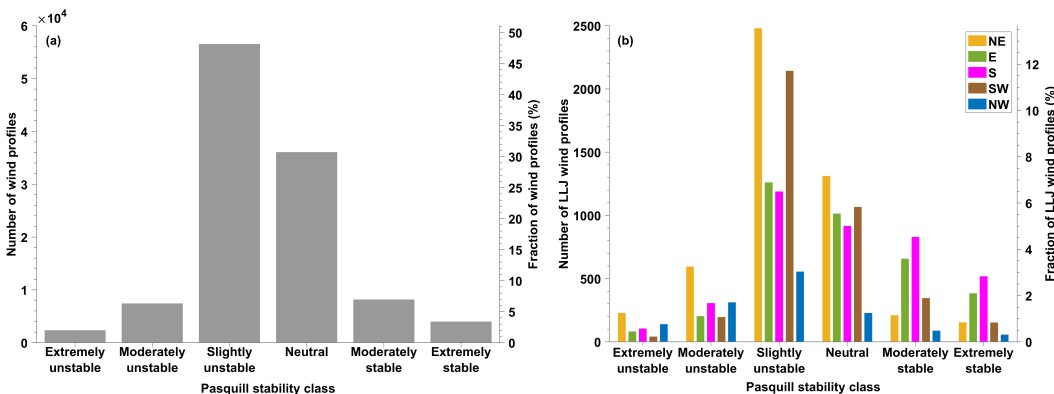

**Figure 10.** Distribution of the Pasquill atmospheric stability classes for (a) all wind profiles and (b) low-level jet profiles, detailed by direction classes, as defined in Sect. 3.2.1.

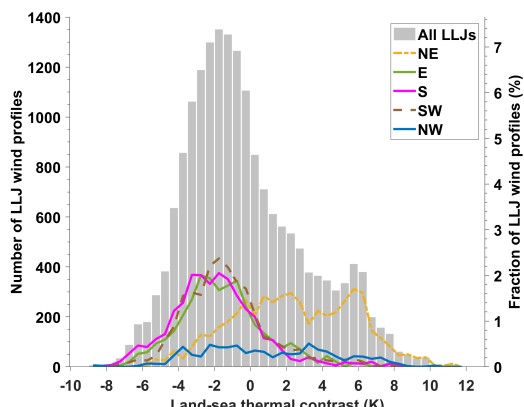

**Figure 11.** Distribution of the land-sea temperature gradient, detailed by the direction classes defined in Sect. 3.2.1, and computed as explained in Sect. 2.3. Fractions indicated on the right y-axis are calculated as a proportion of the total number of LLJ profiles (18,294).

were more frequently observed for negative thermal contrasts, ranging between -8 K and 2 K, as these jets were predominantly nocturnal.

## 3.4 Impact on wind turbines

### 3.4.1 Power production

Figure 12a presents the distribution of the rotor-averaged wind speed for LLJ and non-LLJ profiles for conventional, advanced and future wind turbines. As expected, larger wind turbines would have intercepted higher winds, whether there was a jet or not, indicating that larger turbine dimensions are advantageous for increasing energy production. Indeed, for non-LLJ profiles,

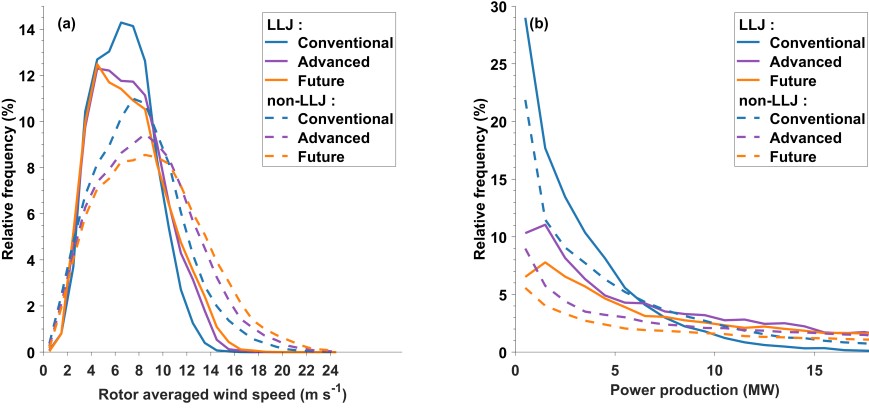

**Figure 12.** (a) Distribution of the rotor-averaged wind speed for low-level jet and non-low-level jet profiles and for conventional, advanced and future wind turbines. (b) Distribution of the power production of the three wind turbine types under low-level jet and non-low-level jet conditions. The y-axes represent the frequency of occurrence normalized by the total number of LLJ profiles (18,294) or non-LLJ profiles (99,117).

the mean rotor-averaged wind speed increased from $8.0\,\mathrm{m\,s^{-1}}$ for conventional turbines to $8.7\,\mathrm{m\,s^{-1}}$ for advanced turbines, and further to $9.1\,\mathrm{m\,s^{-1}}$ for future turbines, representing an increase of approximately 14 % from conventional to future
turbines. However, the equivalent wind speed distribution within the rotor layer for LLJ profiles tended to be centred around lower wind speed values compared to non-LLJ profiles, with mean wind speeds of $6.7\,\mathrm{m\,s^{-1}}$, $7.1\,\mathrm{m\,s^{-1}}$, and $7.2\,\mathrm{m\,s^{-1}}$ for the conventional, advanced, and future turbines, respectively. This indicates that under LLJ conditions, the mean equivalent wind speed was lower by about 16 % for conventional turbines, 18 % for advanced turbines, and 21 % for future turbines compared to non-LLJ conditions. Figure 12b presents the distribution of the power production for the three types of turbines under LLJ
and non-LLJ conditions. For conventional turbines, jets would have increased the power output at low wind speeds (below $6\,\mathrm{m\,s^{-1}}$), while other extreme conditions would have been more advantageous at higher speeds. On the contrary, LLJs would have been beneficial in all conditions for advanced and future wind turbines.

### 3.4.2 Wind speed shear

Figure 13 shows the distribution of the relative distance parameter $\xi$ between the turbine hub and the jet core, for the three types
of wind turbines. The results show that current wind turbines would mainly have experienced entirely positive shear during LLJ events, with $\xi \leq -1$ for 56 % of the LLJ profiles. This fraction would have been reduced to 21 % for future turbines. In agreement with the LLJ core height distribution (Sect. 3.2.3), an important fraction of jet cores, from 44 % to 79 %, would have been located within the rotor area of all turbines, resulting in a combination of positive and negative shear on the blades ($-1 < \xi < 1$). This proportion logically becomes even more important as the turbine dimensions grow. Although jet cores



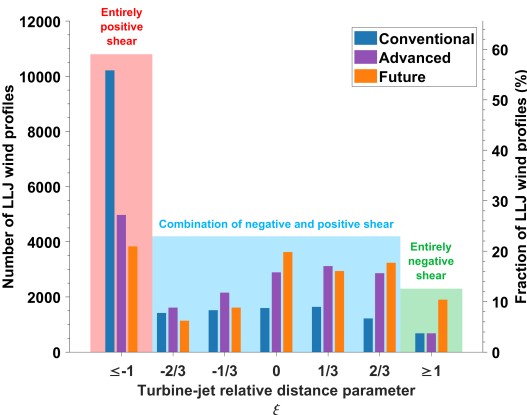

**Figure 13.** Distribution of the turbine hub-jet core relative distance parameter for all the low-level jet profiles. The red zone corresponds to entirely positive shear ($\xi \leq -1$), the blue zone to a combination of positive and negative shear ($-1 < \xi < 1$) and the green zone to entirely negative shear across the rotor ($\xi \geq 1$). Fractions indicated on the right y-axis are calculated as a proportion of the total number of LLJ profiles (18,294).

located below the rotor zone were less frequent, future turbines would have encountered such cases approximately 2.7 times more often than current turbines. Consequently, the proportion of negative shear will increase with larger turbine sizes.

Figure 14 presents in more detail the shear distribution in the rotor area, from the upper and lower blade tips to the hub, for current and future wind turbines under LLJ and non-LLJ conditions. Since the results obtained for the intermediate turbine are similar to those obtained for future turbines, they are not presented here. Figure 14a indicates that the shear distribution in the rotor area of conventional turbines was globally similar in jet and non-jet conditions. For future wind turbines experiencing LLJs, a shift toward negative shear values was observed on the upper blade (Fig. 14b), going from a mean value of $0.0084 \text{ s}^{-1}$ for smaller turbines to a mean value of $-0.0041 \text{ s}^{-1}$ for future turbines. On the contrary, a slight shift toward positive shear values was observed on the lower blade, indicating that LLJ cores would have mainly been located in the hub zone, inducing higher winds at the blade tips.

### 3.4.3 Wind direction shear

Figure 15 shows the distributions of the absolute directional shear for conventional, advanced and future wind turbines for LLJ and non-LLJ conditions. In both cases, future wind turbines, operating at higher altitudes, would have been exposed to lower directional shear than current wind turbines. Moreover, this directional shear across the rotor layer was more pronounced during LLJ events, as the distribution shows a higher proportion of larger values in these cases. The magnitude of the mean directional shear reached a value of $0.125 \text{ °m}^{-1}$ for LLJ profiles, compared to $0.090 \text{ °m}^{-1}$ for other cases, corresponding to a 38 % increase for conventional turbines. Future turbines also showed an important increase, with the mean absolute directional



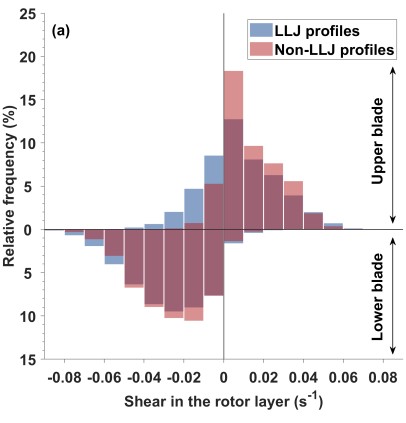
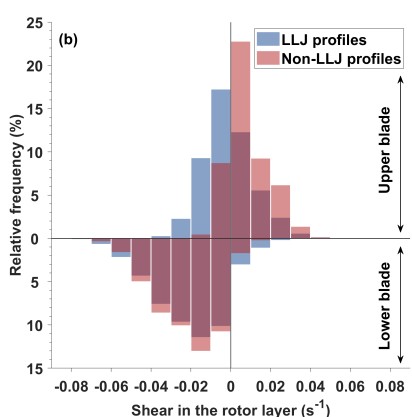

**Figure 14.** Shear distribution in the rotor area from the upper and lower blade tips to the hub (upper and lower sides of the y-axis, respectively) for (a) conventional and (b) future wind turbines. The blue and red areas represent low-level jet and non-low-level jet profiles, respectively, with overlapping regions shown in dark red. The y-axes represent the frequency of occurrence normalized by the total number of LLJ profiles in the rotor layer or non-LLJ profiles.

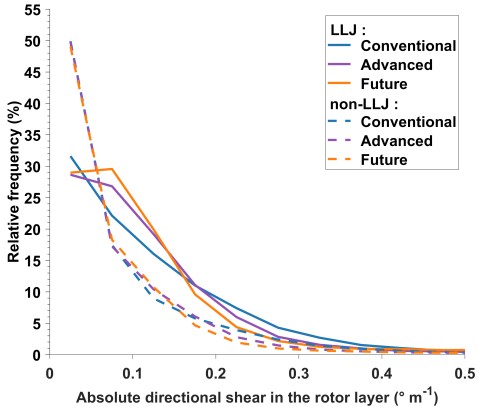

**Figure 15.** Distribution of the absolute directional wind shear in the rotor layer of conventional, advanced and future wind turbines, under low-level jet and non-low-level jet conditions. The y-axis represents the frequency of occurrence normalized by the total number of LLJ profiles (18,294) or non-LLJ profiles (99,117).

shear going from $0.068~°\mathrm{m}^{-1}$ in non-LLJ conditions to $0.105~°\mathrm{m}^{-1}$ under LLJ conditions, corresponding to an increase of $54~\%$.





## 4 Discussion

In this section, results on the jets' characteristics will be compared to other studies carried out in the North Sea region. These studies include the inland locations of Cabauw (Baas et al., 2009), Braunschweig (Ziemann et al., 2020) and WiValdi (Wildmann et al., 2022), the offshore sites of IJmuiden (Kalverla et al., 2017; Duncan, 2018; Kalverla et al., 2019), FINO1 (Wagner et al., 2019) and Heligoland (Rausch et al., 2022), and the coastal locations of Norderney (Rausch et al., 2022) and Dunkerque (Dieudonné et al., 2023). The locations of these sites are shown in Fig. 1a.

### 4.1 Jet characteristics

#### 4.1.1 Core direction

The distribution of the jet core directions (Fig. 4b) is comparable to the one obtained previously in Dunkerque by Dieudonné et al. (2023), with the dominant directional group corresponding to northeasterly jets. However, southwesterly jets were nearly absent in the previous study, which was limited to about 300 m in height, indicating that southwesterly jets occurred at higher

altitudes. This result can also be observed from the distribution of LLJ core heights detailed by core direction (Fig. 6a) and is confirmed by Duncan (2018) and Wagner et al. (2019), who highlighted that southwesterly jets tended to be located at higher altitudes at the IJmuiden and FINO1 sites.

#### 4.1.2 Core speed

The distribution of the jet core speeds above Dunkerque (Fig. 5) is similar to those reported at the inland locations of Cabauw,

Braunschweig and WiValdi (Baas et al., 2009; Ziemann et al., 2020; Wildmann et al., 2022), with jet speeds most frequently observed between 6 and 10 m s$^{-1}$. However, in contrast to these studies, jet speeds between 4 and 6 m s$^{-1}$ were also frequent in Dunkerque, which may result from the presence of weaker jets in coastal areas, particularly sea breezes. This leads to slightly lower average core speeds compared to those found by Duncan (2018) and Wagner et al. (2019) at the offshore sites of IJmuiden and FINO1, but slightly higher than what was reported in the previous study by Dieudonné et al. (2023), likely due

to the contribution of higher-altitude jets.

#### 4.1.3 Core altitude

The altitude distribution (Fig. 6) differs from the previous observations in Dunkerque (Dieudonné et al., 2023), which showed a decrease in jet frequency above 200 m. This discrepancy is attributed to both the reduced data availability at these altitudes and the measurement limit of 300 m, which prevented the reliable detection of jets up to this height. On the contrary, the most

frequent altitude range for the jets in Dunkerque (100 to 200 m) was in agreement with the findings of Rausch et al. (2022) at the coastal site of Norderney. The LLJ core heights in coastal zones appear to be intermediate between the slightly lower-altitude jets observed offshore (Kalverla et al., 2017; Duncan, 2018; Kalverla et al., 2019) and the higher-altitude jets more commonly found inland (Baas et al., 2009; Ziemann et al., 2020; Wildmann et al., 2022). The relationship between altitude and





jet core speed shows the presence of weaker jets at low levels and stronger jets at higher altitudes, which is consistent with the

tendency for LLJ core speeds to increase with height as reported in other studies (Banta et al., 2002; Baas et al., 2009; Ziemann et al., 2020; Rausch et al., 2022). Nevertheless, the high-altitude LLJs observed in this study were likely underestimated due to the decrease in data availability with altitude, as in all ground-based remote sensing studies.

### 4.1.4 Annual cycle

Jets represented 15.6 % of the profiles measured over the study period. However, this percentage varied between 11.1 % and

20.3 % when considering the years individually (Table A2). These differences can be explained by the interannual variability in the weather conditions, though the varying data availability also played a role, with the months experiencing high or low jet frequencies being sampled differently over the successive years. The annual cycle (Fig. 7) is coherent with the observations from the sites of Cabauw, IJmuiden, Heligoland, Norderney and with prior results from Dunkerque (Baas et al., 2009; Kalverla et al., 2017; Duncan, 2018; Rausch et al., 2022; Dieudonné et al., 2023), who also reported higher LLJ occurrences in spring

and summer, fewer in autumn, and even less in winter. The peak of LLJ frequency in June is similar to the one reported by Kalverla et al. (2019), while it differs from the maximum occurrence in April found by Dieudonné et al. (2023). This difference is attributable to the strong contribution of southwesterly jets to this peak, while these jets were not observed previously in Dunkerque.

### 4.1.5 Diurnal cycle

In Dunkerque, LLJs were more frequent at night, which is a feature commonly reported by other studies (Baas et al., 2009; Duncan, 2018; Wagner et al., 2019; Rausch et al., 2022). This diurnal cycle (Fig. 8) differs from the one reported by Dieudonné et al. (2023), who observed two maxima of similar amplitude in the night and in the early afternoon. This suggests that a significant part of nighttime jets was located beyond the range covered by the short-range Doppler lidar, likely the southwesterly jets, that were dominantly nocturnal (Fig. 4d) and occurred at higher altitudes than the others (Fig. 6a). Nevertheless, both

studies in Dunkerque reported a morning minimum, in line with the observations of Rausch et al. (2022) at the coastal site of Norderney. In their analysis, they identified the afternoon jets as sea breezes, as afternoon jets were less frequent at their second measurement site located offshore (Heligoland). This interpretation is supported by observations at both inland and offshore locations, where jet occurrence reaches a minimum during the afternoon (Wagner et al., 2019). At coastal sites, the observed morning plateau with low jet frequency corresponds to a transitional phase between the dissipation of nocturnal jets and the

onset of sea breezes.

### 4.1.6 Event duration

The duration of jet events was not investigated in the North Sea, apart from the work of Dieudonné et al. (2023), who found an average duration of 3.1 h, which is shorter than the 5.4 h obtained in the present study. This difference may be attributed to the use of different continuity criteria. As a much shorter threshold was used in the prior study (20 min instead of 1.5 hour),





shorter jet events were included, which lowered their average duration. Moreover, the limited observation range may result in missing part of a jet event (when the core rises above the detection limit), which may have impacted the duration of the events in the prior study.

## 4.2 Formation mechanisms

The main characteristics of LLJs depending on their direction, that were highlighted in Sect. 3.2 and 3.3, enable the identifica-
tion of their possible formation mechanisms. Northeasterly LLJs came from the sea and formed predominantly during the day, with a frequency peaking in the afternoon. These jets were mainly observed during summer, and were almost absent in winter. They occurred most frequently when the land was warmer than the sea, under relatively unstable atmospheric conditions. All these findings indicate that these jets may correspond to sea breezes. However, the presence of both high-velocity and weaker jets in this group, along with their persistence into the late afternoon and evening, suggests that they likely resulted from the
combination of two mechanisms. Their direction, almost parallel to the coastline, implies that the second mechanism contributing to their development may have been the channeling of continental air masses into the Dover Strait under anticyclonic conditions. The jet formation process in this case has been related to the orography and land-sea roughness contrast (Capon, 2003).

Similar characteristics were observed for northwesterly LLJs. These jets generally presented lower velocities, occurred most
frequently during summer afternoons, and were absent in winter, which suggests that they were mostly composed of sea breezes. Their direction being perpendicular to the coastline, these jets would correspond to purely local sea breeze cases, without contribution of the wind channeling into the Dover Strait. However, northwesterly LLJs also developed under negative land-sea temperature gradients, could occur in the late night, and were observed for all stability classes, indicating that a secondary mechanism likely contributed to their formation.

Easterly and southerly LLJs presented rather similar characteristics, as both mainly occurred at night and under stable atmospheric conditions, which points towards the frictional decoupling formation mechanism. Additionally, the offshore direction of these jets, their occurrence associated with a sea warmer than the land, and the higher frequency of easterly jets in spring and summer imply a possible contribution of land breezes.

Southwesterly LLJs were mainly nocturnal, but were very rare under stable atmospheric conditions, so that frictional decou-
pling is not a likely formation mechanism for these jets. However, southwesterly LLJs could be associated with the passage of cold fronts, or result from the channeling of oceanic air masses in the Dover Strait under cyclonic conditions.

## 4.3 Impact on wind turbines

Using the entire wind profile over the rotor disk, rather than a single measurement at the hub height, provided a more accurate estimation of the power output of the turbines, especially in the context of LLJ profiles and larger rotor dimensions. For
conventional wind turbines, the specific shape of LLJ profiles – with high wind speeds limited to a narrow strip around the jet core and rapidly decreasing above and below – resulted in lower rotor-averaged wind speeds compared to other strong wind conditions. However, because LLJ core speeds tend to increase with height, LLJ will improve the power output of advanced



and future turbines. This conclusion is consistent with the findings of Weide Luiz and Fiedler (2022), who reported that the conventional turbine average energy production increased during nocturnal LLJ events, with the effect being even more

advantageous for larger turbines.

LLJ-induced wind shear can have both negative and positive impacts on turbines, depending on the location of the jet core relative to the turbine hub. For conventional turbines, an important part of the jets was located within the rotor area (Fig. 13) and in this zone, the shear was comparable between LLJ and non-LLJ conditions (Fig. 14a). Indeed, conventional turbines are most often situated within the surface layer, where shear is predominantly influenced by ground effects rather than by the

presence of a jet. Besides, as most jets were located at low altitudes (Fig. 6), implementing taller wind turbines will allow for an increasing proportion of jet cores to be located below the altitude band swept by the rotor. This configuration is the most beneficial for the turbines as it reduces the loads experienced by the nacelle and tower (Gutierrez et al., 2017).

As for wind speed shear, the directional shear has a non-negligible impact on wind turbines, as it causes asymmetry in turbine wakes, which can affect downstream turbines (Abkar and Porté-Agel, 2016). These wake distortions are more pronounced for

larger directional shears, and can alter the distribution of forces on turbine blades, impacting mechanical loads and potentially increasing wear (Robertson et al., 2019; Kapoor et al., 2020). Additionally, directional shear also impacts turbine performance, as large values can result in a decrease in power output (Sanchez Gomez and Lundquist, 2020). LLJs in Dunkerque were found to increase directional shear across the rotor area for all turbine sizes, which is coherent with the findings of Weide Luiz and Fiedler (2022), who reported an intensification of the directional shear under nocturnal LLJ conditions. However, the results

indicate that this effect will be less pronounced for future turbines operating at higher altitudes, since the impact of ground friction on wind direction shear decreases with height (Tumenbayar and Ko, 2023).

## 5   Conclusions

This study analysed the characteristics of LLJs up to 1,500 m in Dunkerque, a French coastal city in the southern part of the North Sea. The main properties of LLJs were determined using 3.3 years of observations, with wind profiles recorded by two

long-range Doppler lidars. The impact of these jets was assessed on three hypothetical wind turbines of varying sizes, in terms of power production, as well as vertical shear in wind speed and direction, both of which are known to affect turbine loads.

LLJs were found to be a non-negligible phenomenon since they occurred 15.6 % of the time. The mean core speed was 8.4 $\mathrm{m\,s^{-1}}$, and the jet cores were on average located at a 267-m altitude. The LLJs presented a pronounced annual cycle, revealing a maximum occurrence in the spring and summer seasons and a minimum during the cold months. Similarly, their

occurrence varied according to a marked daily cycle, the jets being more frequent at night, although an increase was also noted in the afternoon, a feature commonly observed at coastal sites.

LLJs were categorized into major directional groups in order to identify their formation mechanisms. Northeasterly LLJs, which mainly occurred in the afternoon when the land was warmer than the sea, were probably a combination of sea breezes and wind channeling in the Dover Strait. For the same reasons, a part of the daytime northwesterly jets likely corresponded to

pure sea breezes. Easterly and southerly LLJs occurred predominantly at night, when the sea was warmer than the land, and





with a preference for stable conditions, which suggests they resulted from frictional decoupling and land breezes. The direction of southwesterly jets, and the fact that they were not associated with stable conditions, suggests they more likely resulted from wind channeling in the Dover Strait and from frontal passages.

LLJ cores were located within the rotor area of current turbines for 44 % of the cases, and even 79 % concerning future turbines, which represents a steady 12.3 % of the time. Unexpectedly, LLJs were associated with lower rotor-averaged wind speeds compared to non-jet situations, which can be explained by the fact that the wind enhancement associated with the jet is limited to a narrow strip not covering the full rotor area. For the current, smaller wind turbines, the resulting energy production was thus higher under non-jet conditions at high wind speeds. For future and larger wind turbines, however, LLJs will be beneficial for energy production under all conditions.

The high vertical extent of the measurements revealed that higher-altitude jets also impact turbines, notably through the shear they induce. In the presence of a jet, current wind turbines were found to be mostly exposed to shear conditions that increase both static and dynamic loads on their structure, but the growth in size of future wind turbines was demonstrated to reduce these negative impacts. An analysis separating shear on the lower and upper blades revealed that current turbines experienced similar shear whether there is a jet or not, while for future and taller turbines, the shear will be reduced by the presence of a jet. Directional shear was observed to be less pronounced for taller turbines, whether under jet or non-jet conditions. However, directional shear was greater during LLJ events for all turbine sizes.

To further investigate the impact of LLJs on the structure of wind turbines, a study is being conducted in collaboration with structural mechanics specialists, in the framework of the SALOME project (https://www.salome-interreg.eu/). To relate atmospheric conditions to structural response, wind and turbulence profiles will be measured simultaneously with strain measurements on turbines. Aeroelastic simulations will also be performed using the measured wind profiles as input. Regarding LLJ properties, the objective is now to assess the horizontal extent of the different types of jets in the region, by running numerical simulations with the Weather Research and Forecasting (WRF) model. The model's ability to accurately reproduce the different types of jets will also be evaluated by comparison with the lidar wind profiles, for different planetary boundary layer parametrizations. These simulations will ultimately provide a better understanding of the different formation mechanisms of LLJs, as well as the way they form and dissipate.





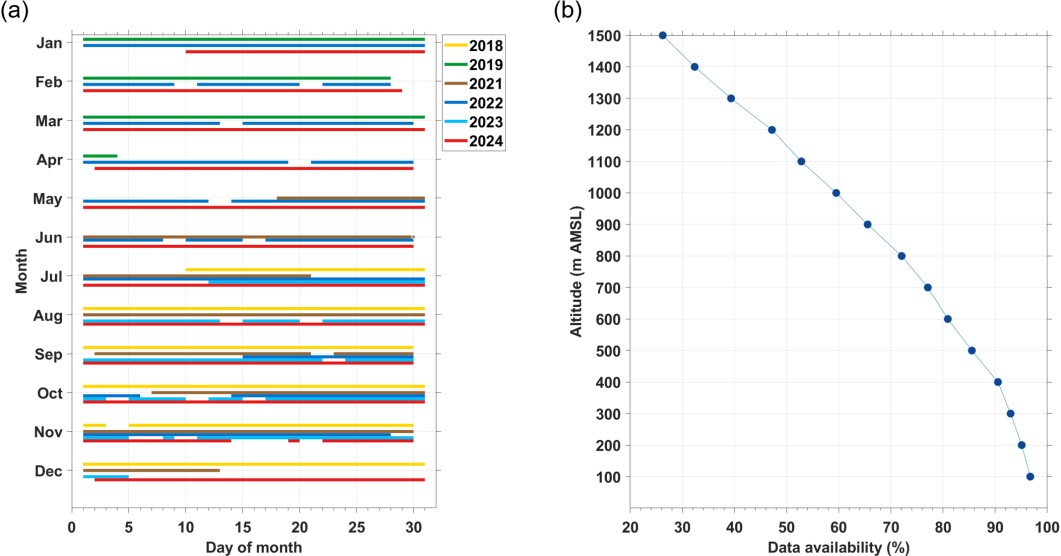

**Figure A1.** Data availability (a) for each year and month covered by the study and (b) as a function of the altitude of the observations.

**Table A1.** Monthly number and fraction of observed wind profiles showing a low-level jet, before and after data availability correction.

| Month | Data availability (%) | Number of profiles with LLJ | | Fraction of profiles with LLJ (%) | |
|---|---|---|---|---|---|
| | | Raw | Corrected | Raw | Corrected |
| January | 45.2 | 615 | 1,362 | 3.36 | 3.84 |
| February | 48.8 | 703 | 1,440 | 3.84 | 4.06 |
| March | 48.9 | 1,780 | 3,638 | 9.73 | 10.2 |
| April | 34.4 | 1,111 | 3,226 | 6.07 | 9.10 |
| May | 40.3 | 1,477 | 3,663 | 8.08 | 10.3 |
| June | 49.4 | 2,500 | 5,056 | 13.7 | 14.3 |
| July | 60.1 | 2,878 | 4,789 | 15.7 | 13.5 |
| August | 56.2 | 2,174 | 3,867 | 11.9 | 10.9 |
| September | 63.3 | 2,618 | 4,134 | 14.3 | 11.7 |
| October | 64.5 | 1,677 | 2,595 | 9.17 | 7.33 |
| November | 66.2 | 449 | 678 | 2.45 | 1.91 |
| December | 30.7 | 312 | 1,018 | 1.70 | 2.86 |
| Average / Total | 50.7 | 18,294 | 35,466 | 100 | 100 |

# Appendix A: Data availability



**Table A2.** Number and fraction of wind profiles showing a low-level jet for each year of observation.

| Year | 2018 | 2019 | 2021 | 2022 | 2023 | 2024 |
|---|---|---|---|---|---|---|
| Number of profiles with LLJ | 2,686 | 1,218 | 4,006 | 5,533 | 1,547 | 3,304 |
| Fraction of profiles with LLJ | 15.1% | 12.7% | 20.3% | 19.9% | 12.3% | 11.1% |

*Code and data availability.* The code developed and used for this study can be made available by the corresponding author upon request. The wind lidar data and the anemometer data are available from the corresponding author upon request. The raw ERA5 data are available from the Climate Data Store of the Copernicus Climate Change Service (https://cds.climate.copernicus.eu/). The data derived from the reanalyses

are available from the corresponding author upon request.

*Author contributions.* PH, ED, AS and HD conceptualised this study and developed the methodology. PA and MF installed and monitored the instruments. PH processed the data, analysed the results and prepared the initial draft of the paper. All authors participated in reviewing and editing the manuscript.

*Competing interests.* The authors declare that they have no conflict of interest.

*Acknowledgements.* The authors gratefully acknowledge the financial support of the Pôle Métropolitain de la Côte d'Opale (PMCO) and the Région Hauts-de-France. The authors also thank the Halle aux Sucres Learning Centre for hosting the lidar instruments and the ECMWF for providing the ERA5 data. The statistical study presented in this work was carried out using the CALCULCO computing platform, supported by SCoSI ULCO (Service Commun du Système d'Information de l'Université du Littoral Côte d'Opale).

*Financial support.* This project is co-funded by the Pôle Métropolitain de la Côte d'Opale (PMCO) and by the Région Hauts-de-France. The

authors gratefully acknowledge the financial support received by the Interreg VI France-Wallonie-Vlaanderen programme, co-funded by the European Union, under SALOME project. The lidar instruments were funded through the CPER IRenE by the Région Hauts-de-France, the ERDF and the CPER ECRIN.



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
