# Peer review of "Low-level jets in the southern North Sea: implications for wind turbine performance using Doppler lidar observations"

_Wind Energy Science, 2025_

## Referee Comment (RC1)

Review of the manuscript wes-2025-183

Low-level jets in the southern North Sea: implications for wind turbine performance using Doppler lidar observations

By Pauline Haezebrouck and co-workers

Summary: This manuscript reports on a study using a Doppler lidar in Dunquerke at the French North Sea coast to develop a mini-climatology (3.3 years) of low-level jets (core speed, height, shear, Pasquill class) in the context of wind energy studies. The paper is in general well written, with clear figures, and is put appropriately in the context of earlier studies in the study area. Despite the figures are clear, with 15 figures they are with many. The main new insight that is provided is that based on the wind speed profile climatology with turbines of the future will suffer less from extreme positive wind shears in the LLJ-profiles, since they will "grow beyond them". More work is needed on the manuscript on the aspects of the understanding of the LLJ mechanisms.

Recommendation: Major revision.

**Major remarks:**

- -Although the sec derivation of the LLJ climatology from the Doppler lidar data is thoroughly done, the paper is relatively silent about the challenging measurement site, i.e. close to the coast. The paper compares their findings with sites at FINO and Ijmuiden which is over sea, and Cabauw which truly over land. However the Dunquerke site is at the coast and the sensors will feel influences from land and sea depending on the wind direction and trajectories of the airmasses. This hampers the interpretation of LLJ climatology and should be addressed more in the paper. Some more concrete concerns are in the minor remarks below.
- -Section 4.2 should be strengthened. This section provides physical explanations for the LLJ climatology, but many of them are suggestions (plausible suggestions, I agree), but I would like to see more evidence for the statements that are made in section 4.2 on the LLJ groups at high and low elevations, and the LLJ formation mechanism in the different wind directions.
- -The paper has a relatively large amount of figures, and some are discussed not in much detail, so please consider ad deeper explanation and/or merging some figures.

Minor remarks:

Ln 90: To address this issue, the number of LLJ wind profiles was adjusted by dividing the observed values by the corresponding fraction of available data. This is a bit risky since the synoptic patterns might not be the same in the sample set. Do your results remain the same if the number of LLJs is not scaled with time/available data, but by available data per weather pattern (Lamb weather type, GrossWetterlagen)?

Figure 2: please add to the x axis it represents wind speed and to the y axis it represents height.

Ln 112: In this study, a longer criterion was considered as it improved the jet detection by excluding isolated jets. Please justify how you measured it was a "improved jet detection". What is your ground

truth? Also, isolated jets are also jets, and they can affect wind turbines, and as such should be included in the statistics.

Ln 119: please add which value of the Von Karman constant was used. I have seen values between 0.35-0.41 in the literature. Just for reproducibility reasons.

Ln 119: turbulent sensible heat flux: at which height. I suggest to reword it to turbulent sensible heat flux at the surface (if it is sufficiently close to justify this). Same for friction velocity.

Equation 1: The Greek capital L to denote the Obukhov length suggests that this Obukhov length is the local Obukhov length that is based on local fluxes at height z (see Nieuwstadt, 1984). If you intend to use the Obukhov length based on surface fluxes, then I suggest to move to regular capital L for its notation.

Ln 126: I would like to see a sentence or two here that says something about the quality of ERA5 "to derive the conditions favoring the formation of LLJs". What is the quality of this product for your study area? The ERA5 product has a good reputation, but is not perfect in LLJs (see e.g. https://rmets.onlinelibrary.wiley.com/doi/10.1002/qj.3748). Also, ERA5 is relatively coarse with 0.25 deg grid spacing. Since Dunquerke is at the coast of the British Channel, is the ERA5 gridcell you used a land grid cell or sea grid cell? This will make a big difference and need to be addressed.

Ln 144: The assumption of the air density being constant at 1.2 kg/m3 seems to me as a very crude assumption, since others use 1 kg/m3 as well. For didactic reasonings it is okay, but here you aim to estimate real production values, so a density of 1 vs a density of 1.2 means 20% difference/bias/error in the estimate of P. So it is better to calculate and use the actual density.

Ln 206: It okay to report the mean LLJ core speed. But perhaps also add the median if the distribution is not very close to a normal distribution.

Ln 230: around this line you discuss the climatology of the fast jets being at higher elevation, and weaker jet being closer to the surface/lower elevation. Could you link this to their mechanism of formation? I can imagine the lower jets to be triggered by the land/sea contrast in surface roughness and/or surface temperature difference, rather than by the traditional decoupling in stable boundary layer conditions. This would bring added value to the manuscript.

Section 3.3.1: it is surprising that so many LLJs occur under Unstable condition, since the theory says LLJs many occur under stable conditions after decoupling of the atmosphere from the land surface, and a subsequent inertial oscillation that starts. So what is the quality of if your Pasquill classes categorisation? The sign of L will largely depend on the footprint of the sonic anemometer, which is located at the coast. So with wind directions between NE (45 deg )and SW (225 deg) it feels land in the footprint and sea otherwise (roughly). But the sign of the surface fluxes can depend very much on this footprint, and as such the Pasquill class an event gets assigned to. All of this should be clarified in a revised version of the paper.

Section 3.3.1: in addition, to what extent is it possible to quantify whether the detected LLJs have been produced/triggered elsewhere (e.g. over the North Sea) and were just advected to your study site. It would be interesting to have this information to clarify the LLJ statistics behavior.

Ln 400: "implies that the second mechanism contributing to their development may have been the channeling of continental air masses into the Dover Strait under anticyclonic conditions. The jet formation process in this case has been related to the orography and land-sea roughness contrast". I agree that there must be multiple processes active, but more justification is needed for this statement. Running an illustrative case like this with a high resolution weather forecasting model like WRF would help to secure the understanding and proposed mechanism is correct.

Ln 404-: These jets generally presented lower velocities, occurred most frequently during summer afternoons, and were absent in winter, which suggests that they were mostly composed of sea breezes. Here I would disagree with the statement (or at least not enough support is given to justify the statement), since the current paper does not make a climatology of sea breezes to link the two phenomena. It can also be that in summer the North Sea SST is relatively low compared to its overlying atmosphere, which promotes a stable boundary layer, decoupling of the atmosphere from the surface and inertial oscillations can trigger LLJs. If these are then advected to your study site with a NW wind,

| they will not be triggered by a sea breeze. I also say this since my experience says the amount of pure sea breezes over the North Sea coast is less than our intuition might say. |
|------------------------------------------------------------------------------------------------------------------------------------------------------------------------------------|
|                                                                                                                                                                                    |
|                                                                                                                                                                                    |
|                                                                                                                                                                                    |
|                                                                                                                                                                                    |
|                                                                                                                                                                                    |
|                                                                                                                                                                                    |
|                                                                                                                                                                                    |
|                                                                                                                                                                                    |
|                                                                                                                                                                                    |
|                                                                                                                                                                                    |
|                                                                                                                                                                                    |
|                                                                                                                                                                                    |
|                                                                                                                                                                                    |
|                                                                                                                                                                                    |
|                                                                                                                                                                                    |
|                                                                                                                                                                                    |
|                                                                                                                                                                                    |
|                                                                                                                                                                                    |

---

## Author Comment (AC1)

**Authors' response to reviewers' comments on article:**

**"Low-level jets in the southern North Sea: implications for wind turbine performance using Doppler lidar observations" (Manuscript 2025-183).**

First, we want to thank the reviewers for their careful reading of the manuscript and their constructive comments, which have helped to improve the quality of this paper. We have addressed all the remarks, and the manuscript has been revised accordingly. Below you can find detailed answers to all your comments.

**Reviewer 1:**

**Summary:**

This manuscript reports on a study using a Doppler lidar in Dunquerke at the French North Sea coast to develop a mini-climatology (3.3 years) of low-level jets (core speed, height, shear, Pasquill class) in the context of wind energy studies. The paper is in general well written, with clear figures, and is put appropriately in the context of earlier studies in the study area. Despite the figures are clear, with 15 figures they are with many. The main new insight that is provided is that based on the wind speed profile climatology with turbines of the future will suffer less from extreme positive wind shears in the LLJ-profiles, since they will "grow beyond them". More work is needed on the manuscript on the aspects of the understanding of the LLJ mechanisms.

**Recommendation:** Major revision.

**Major remarks:**

- Although the sec derivation of the LLJ climatology from the Doppler lidar data is thoroughly done, the paper is relatively silent about the challenging measurement site, i.e. close to the coast. The paper compares their findings with sites at FINO and Ijmuiden which is over sea, and Cabauw which truly over land. However the Dunquerke site is at the coast and the sensors will feel influences from land and sea depending on the wind direction and trajectories of the airmasses. This hampers the interpretation of LLJ climatology and should be addressed more in the paper. Some more concrete concerns are in the minor remarks below.

We acknowledge that the coastal location of the Dunkerque site presents challenges for the interpretation of the climatology of the LJJs, as the measurements can be influenced by both land and sea depending on the wind direction. This is indeed a specificity of our site compared to the purely offshore or inland sites mentioned in our comparison. This question was already addressed in a previous work from Maynard et al. (2025) in Dunkerque, who studied the turbulent coherent structures in the surface layer using Doppler lidar horizontal sweeps. The results showed that, in the instrument range, onshore air masses kept their turbulent characteristics when advected seaward, while offshore air masses rapidly transitioned to onshore turbulence characteristics within a few hundred meters of the coast. This rapid adjustment indicates that the measurements from both the ultrasonic anemometer and the Doppler lidars mostly capture onshore atmospheric conditions, even for sea wind sectors. We think that this supports the validity of our LLJ climatology interpretation.

- Section 4.2 should be strengthened. This section provides physical explanations for the LLJ climatology, but many of them are suggestions (plausible suggestions, I agree), but I would like to see more evidence for the statements that are made in section 4.2 on the LLJ groups at high and low elevations, and the LLJ formation mechanism in the different wind directions.

To support our suggestions for the formation mechanisms for the different LLJ wind directions, we added references to studies from the North Sea sites that reported similar LLJ characteristics and

attributed them to specific formation mechanisms. These include frictional decoupling for nocturnal jets (Baas et al., 2009; Ziemann et al., 2020; Wildmann et al., 2022), which correspond to easterly and southerly LLJs in Dunkerque and sea breezes (northwesterly and northeasterly jets) for coastal locations (Rausch et al., 2022). We also added references to Miller et al. (2003) and Steele et al. (2015) regarding the orientation of sea breeze fronts relative to the coastline, which enabled us to discuss the occurrence of coastal jets associated with corkscrew sea breezes (northeasterly jets). Finally, as suggested in the minor remarks, we expanded our discussion to include LLJs that form over the sea through frictional decoupling and are advected onshore to Dunkerque.

- The paper has a relatively large amount of figures, and some are discussed not in much detail, so please consider a deeper explanation and/or merging some figures.

Figures 10a and 10b have been merged into a new fig. 10a, and fig. 11 is now fig. 10b. For the rest, merging the subfigures (e.g. fig 6a and 6b) would make them difficult to read.

**Minor remarks:**

Ln 90: To address this issue, the number of LLJ wind profiles was adjusted by dividing the observed values by the corresponding fraction of available data. This is a bit risky since the synoptic patterns might not be the same in the sample set. Do your results remain the same if the number of LLJs is not scaled with time/available data, but by available data per weather pattern (Lamb weather type, GrossWetterlagen)?

Following your remark, we computed the annual cycle by correcting the number of LLJ profiles based on data availability within each Lamb weather type category, and we found similar results, as it is shown on the figure below. Therefore, we kept the original figures in the paper.

**Annual cycle of LLJs**

[Figure]

Figure 2: please add to the x axis it represents wind speed and to the y axis it represents height.

We added these labels to the figure.

Ln 112: In this study, a longer criterion was considered as it improved the jet detection by excluding isolated jets. Please justify how you measured it was a "improved jet detection". What is your ground truth? Also, isolated jets are also jets, and they can affect wind turbines, and as such should be included in the statistics.

The performance of the jet detection process was assessed, for different values of the time-continuity criterion, by comparing the detected jet cores with the lidar time-height cross-sections of the horizontal wind speed, for the entire period of the study. A long continuity criterion of 1.5 h was

selected because we chose to focus on LLJs, which are persistent phenomena, rather than transient extreme wind events, though the latter also affect wind turbines. Nevertheless, to evaluate the influence of these events on our results, we reduced the time continuity criterion from 1.5 hours to 30 minutes and replotted the corresponding curves (figures below). The results are all very similar with both time-continuity criteria, which shows that such phenomena have a negligible impact on our results, with similar curves. This point has been added at the end of the discussion (in section 4.3).

[Figure]

**Turbine hub-jet core relative distance parameter**

[Figure]

[Figure]

**Wind shear in the rotor area of conventional wind turbines**

[Figure]

[Figure]

**Wind shear in the rotor area of future wind turbines**

[Figure]

[Figure]

[Figure]

**Absolute directional wind shear in the rotor layer**

Ln 119: please add which value of the Von Karman constant was used. I have seen values between 0.35-0.41 in the literature. Just for reproducibility reasons.

We detailed the value (κ = 0.37) in the text.

Ln 119: turbulent sensible heat flux: at which height. I suggest to reword it to turbulent sensible heat flux at the surface (if it is sufficiently close to justify this). Same for friction velocity.

Since these parameters are provided by the ultrasonic anemometer at 15 m AMSL, we can consider it to be near the surface. We added this information to the text.

Equation 1: The Greek capital L to denote the Obukhov length suggests that this Obukhov length is the local Obukhov length that is based on local fluxes at height z (see Nieuwstadt, 1984). If you intend to use the Obukhov length based on surface fluxes, then I suggest to move to regular capital L for its notation.

We changed for L instead of Λ.

Ln 126: I would like to see a sentence or two here that says something about the quality of ERA5 "to derive the conditions favoring the formation of LLJs". What is the quality of this product for your study area? The ERA5 product has a good reputation, but is not perfect in LLJs (see e.g. https://rmets.onlinelibrary.wiley.com/doi/10.1002/qj.3748). Also, ERA5 is relatively coarse with 0.25 deg grid spacing. Since Dunquerke is at the coast of the British Channel, is the ERA5 gridcell you used a land grid cell or sea grid cell? This will make a big difference and need to be addressed.

ERA5 was only used to obtain the 2-m temperature at two points (one cell located in the sea, the second one onshore, as shown in the figure below (magenta dots)), in order to compute an estimate of the land-sea temperature gradient during LLJ events. The presence of LLJs is not determined based on ERA5 data, which, as you highlighted, are not so precise in these cases. All the wind data underlying our results came from lidar observations, which provide higher accuracy.

[Figure]

**ERA5 points used to determine the land-sea temperature gradient**

Ln 144: The assumption of the air density being constant at 1.2 kg/m3 seems to me as a very crude assumption, since others use 1 kg/m3 as well. For didactic reasonings it is okay, but here you aim to estimate real production values, so a density of 1 vs a density of 1.2 means 20% difference/bias/error in the estimate of P. So it is better to calculate and use the actual density.

You are right. Instead of computing the power using a constant value of 1.2 kg/m3, we used the air density obtained from measurements of a weather station located near the lidar location (we added the location of the weather station on figure 1b). We replotted the figure accordingly, and we added a description of the weather station in section 2.1.

Ln 206: It is okay to report the mean LLJ core speed. But perhaps also add the median if the distribution is not very close to a normal distribution.

We added the value of the median in the text.

Ln 230: around this line you discuss the climatology of the fast jets being at higher elevation, and weaker jets being closer to the surface/lower elevation. Could you link this to their mechanism of formation? I can imagine the lower jets to be triggered by the land/sea contrast in surface roughness and/or surface temperature difference, rather than by the traditional decoupling in stable boundary layer conditions. This would bring added value to the manuscript.

To investigate the link between the land-sea surface temperature difference and the jet strength and elevation, we replotted Fig. 10b colouring the data by core speed or by core altitude instead of by core direction. The weak jets or the low-altitude jets are not specifically associated with a land warmer than the sea, contrary to the North-East core direction (Fig. 10b). The discussion was made following the core direction groups rather than the core speed, as the core direction appears to be a better discriminator, though still incomplete, between the formation mechanisms.
Regarding decoupling generated by the surface roughness contrast, such a mechanism would form offshore jets, i.e. southerly to easterly jets. There is indeed a significant share of the southerly and easterly jet groups that correspond to weak, low-altitude jets (Fig. 6). However, such jets would form after the air mass has been advected over the sea, so they would not yet exist when the air mass overpasses the lidar. We added a precision about this formation mechanism in Sec. 4.2, in paragraph #3 discussing the southerly and easterly core direction groups.

[Figure]

**Land-sea temperature gradient detailed by LLJ core speed and core height**

Section 3.3.1: it is surprising that so many LLJs occur under Unstable conditions, since the theory says many LLJs occur under stable conditions after decoupling of the atmosphere from the land surface, and a subsequent inertial oscillation that starts. So what is the quality of your Pasquill classes categorisation? The sign of L will largely depend on the footprint of the sonic anemometer, which is located at the coast.

So with wind directions between NE (45 deg )and SW (225 deg) it feels land in the footprint and sea otherwise (roughly). But the sign of the surface fluxes can depend very much on this footprint, and as such the Pasquill class an event gets assigned to. All of this should be clarified in a revised version of the paper.

We agree that the footprint of the sonic anemometer could have impacted the classification into the Pasquill stability classes. This is a question that was already asked in the framework of another of our recent studies, dedicated to the organization of turbulence into coherent structures within the surface layer (Maynard et al., 2025). Horizontal sweeps of the Doppler lidar were recorded from the same site as in this study, and their manual observation showed that, within the range of the lidar, turbulence organization in onshore air masses kept its characteristics while being advected over the sea, while turbulence in offshore air masses transitioned very quickly (within a few hundred meters) to onshore organization. This supports the fact that observations from the ultrasonic anemometer are close to onshore conditions even when the wind comes from the sea, implying that the important part of jets occurring under unstable conditions is not due to an incorrect Pasquill class categorisation. Besides, LLJs occurring under stable conditions and formed by decoupling of the atmosphere from the surface are typically nocturnal, while in Dunkerque, an important proportion (45%) of the LLJs is diurnal and occurs during the warm season, which explains the proportion of the jets falling into the moderately and extremely unstable classes.

Section 3.3.1: in addition, to what extent is it possible to quantify whether the detected LLJs have been produced/triggered elsewhere (e.g. over the North Sea) and were just advected to your study site. It would be interesting to have this information to clarify the LLJ statistics behavior.

It is challenging with a single lidar measurement point to determine whether an observed LLJ is a locally-formed jet or a larger-scale jet advected over Dunkerque. This would require mesoscale simulations to observe the jet onset and its spatial extent. However, the detailed analysis of mesoscale simulation results would concern only a limited number of case studies, while the objective of this paper is to approach LLJs from a statistical perspective. Moreover, preliminary work with the WRF (Weather Research and Forecasting) model shows that even jets that could be considered as local (such as sea breezes) might actually be regional-scale phenomena, for instance, when corkscrew sea breezes first develop across the Belgian coast before extending towards the French coast. Given the scope and the length of the current paper, this question will be addressed in a dedicated future article.

Ln 400: "implies that the second mechanism contributing to their development may have been the channeling of continental air masses into the Dover Strait under anticyclonic conditions. The jet formation process in this case has been related to the orography and land-sea roughness contrast". I agree that there must be multiple processes active, but more justification is needed for this statement. Running an illustrative case like this with a high resolution weather forecasting model like WRF would help to secure the understanding and proposed mechanism is correct.

As said just above, we have started a modelling work using WRF in order to better understand the formation mechanisms and the characteristics of the different types of jets. However, this article is already pretty long, and incorporating another tool will result in increasing the methodology section (to describe the model configuration chosen) as well as the results section (with at least a full-page figure showing wind maps at different times). Moreover, to be balanced, simulations of cases covering the different types of LLJs should be included, which would further increase the amount of content added to the paper and make it overly lengthy.

The mechanisms underlying channeling jets have already been exposed by Capon et al. (2003), as stated in the next sentence of the article. Although this study relied on an old-generation weather model, we think that this reference is sufficient to support the statement that this mechanism contributes to LLJs in Dunkerque. This point is also supported by our first WRF simulations, but as said above, we do not think these results should be included in the present article.

To be clearer about the fact that a single LLJ event can result from a combination of several mechanisms, the sentence in question was modified as: "the second mechanism contributing to their development**, strengthening, or persistence** may have been the channeling of  air masses into the Dover Strait ".

Ln 404-: "These jets generally presented lower velocities, occurred most frequently during summer afternoons, and were absent in winter, which suggests that they were mostly composed of sea breezes." Here I would disagree with the statement (or at least not enough support is given to justify the statement), since the current paper does not make a climatology of sea breezes to link the two phenomena. It can also be that in summer the North Sea SST is relatively low compared to its overlying atmosphere, which promotes a stable boundary layer, decoupling of the atmosphere from the surface and inertial oscillations can trigger LLJs. If these are then advected to your study site with a NW wind,they will not be triggered by a sea breeze. I also say this since my experience says the amount of pure sea breezes over the North Sea coast is less than our intuition might say.

We agree that the mechanism that you describe could also contribute to the northwesterly LLJs observed in Dunkerque and added it to the discussion in Section 4.2 (end of paragraph #2). The beginning of the paragraph was also rephrased for clarity, referring to "pure sea breezes", as named by Miller et al. (2003), while a sentence about corkscrew sea breezes was added in the discussion about northeasterly LLJ (paragraph #1).

**Bibliography:**

- Maynard, P., Dieudonné, E., Sokolov, A., Delbarre, H., Augustin, P., Fourmentin, M., & Dmitriev, E. Exploring the level of organization of turbulent coherent structures in the atmospheric surface layer through supervised classification of Doppler lidar observations. J. Geophys. Res.: Mach. Learn. Comput., 2, e2025JH000652. https://doi.org/10.1029/2025JH000652, 2025.

- Capon, R. A.: Wind speed-up in the Dover Straits with the Met Office new dynamics model, Meteorol. Appl., 10, 229–237, https://doi.org/10.1017/s1350482703003037, 2003.

- Miller, S., Keim, B., Talbot, R., and Mao, H.: Sea breeze: Structure, forecasting, and impacts, Rev. Geophys., 41, https://doi.org/10.1029/2003RG000124, 2003.

- Steele, C., Dorling, S., von Glasow, R., and Bacon, J.: Modelling sea-breeze climatologies and interactions on coasts in the southern North Sea: implications for offshore wind energy, Q. J. R. Meteorol. Soc., 141, 1821–1835, https://doi.org/10.1002/qj.2484, 2015.

- Baas, P., Bosveld, F., Klein Baltink, H., and Holtslag, A.: A climatology of nocturnal low-level jets at Cabauw, J. Appl. Meteorol. Climatol., 48, 1627–1642, https://doi.org/10.1175/2009jamc1965.1, 2009.

- Ziemann, A., Galvez Arboleda, A., and Lampert, A.: Comparison of wind lidar data and numerical simulations of the low-level jet at a grassland site, Energies, 13, 6264, https://doi.org/10.3390/en13236264, 2020.

- Wildmann, N., Hagen, M., and Gerz, T.: Enhanced resource assessment and atmospheric monitoring of the research wind farm WiValdi, J. Phys. Conf. Ser., 2265, 022 029, https://doi.org/10.1088/1742-6596/2265/2/022029, 2022.

- Rausch, T., Cañadillas, B., Hampel, O., Simsek, T., Tayfun, Y. B., Neumann, T., Siedersleben, S., and Lampert, A.: Wind lidar and radiosonde measurements of low-level jets in coastal areas of the German Bight, Atmosphere, 13, 839, https://doi.org/10.3390/atmos13050839, 2022.

**Reviewer 2:**

**Summary:**

The research described in the manuscript investigates low-level jets (LLJs) at Dunkerque, a coastal city on the southern North Sea. The main data were obtained from lidar observations up to an altitude of 1500 m above the ground level. The researchers calculated the atmospheric stability conditions using measurements from a sonic anemometer. They also measured temperatures at two sea points to calculate the land-sea gradient.

Results were compared to several studies in the area, including a similar study previously performed at the same location (Dieudonné et al., 2023). One of the main contributions of the study is that they extended the observational range up to an altitude of 1500 m, indeed multiplying by five the observational range in the previous study (up to 300 m). Consequently, they were able to obtain more accurate results, as they could account for high altitude LLJs previously missing. The authors also used innovative means to compare the impacts on wind turbines of LLJ and non-LLJ winds.

The manuscript presents a topic of high interest and potential practical use, especially to guide the location of future wind energy projects and the sizing of wind turbines. This is particularly important for the region, where energy plays a strategic role. The article is promising; however, there are several concerns that merit the authors attention. I would recommend the authors to address/respond the following comments:

**Major comments:**

1.  The way the authors define and reference the continuity criteria used to select LLJs is not well organized. The authors use two continuity criteria, one in the time scale and another one in the space (altitude) scale, but their uses lack standardization throughout the document.

    Line 109: "… a continuity criterion was introduced …".
    The authors should consider specifying that this is a 'time continuity criterion', to differentiate it from the 'height continuity criterion' loosely described at the end of the same paragraph. If we follow the convention shown in the 1ˢᵗ column of Table 3, the durations of '1.5 hours', '30 minutes', and '1 hour' are not different criteria, but the possible values used for the time continuity criterion.

    To make it clear which type of continuity criterion is referred to, we added the word "time" or "height" throughout the manuscript.

    Lines 113-114: "… LLJ core heights were required not to vary by more than 50 m between two consecutive profiles."
    Since the 2ⁿᵈ column of Table 3 shows not just '50 m' but also '100 m' and 'N/A', this sentence is loosely defining, without naming it, the 'height continuity criterion'. Again, and following the convention shown in the 2ⁿᵈ column of Table 3, the height differences '50 m' and '100 m' are not different criteria, but the possible values used for the height continuity criterion.
    This part has been modified to improve clarity (lines 122-125): "Additionally, **a height continuity criterion was used, with** LLJ core heights required not to vary by more than 50 m between two consecutive profiles. **Sensitivity tests were also performed with different values of the continuity criteria, 30 minutes for time and 100 m or no constraint for height.** The effect of these different **sets of criteria values** on jet detection

is presented in Sect. 3.1."

Since subsection 2.2 is where both continuity criteria are mentioned for the first time in the document, this subsection should explicitly define their names and describe all their possible values.
Lines 172-173: "With the same objective of excluding isolated jets, a continuity criterion in height between two consecutive profiles was introduced." This definition of the height continuity criterion should probably belong to subsection 2.2 (part of the Methodology section), not in the Results section. Here, it will probably suffice to say that the results soon-after correspond to the different values of the height continuity criterion.

We agree that the explanation about multiple jet detection and its relation with the time continuity criterion should have been included in the methodology section. It was therefore moved in Sec. 2.2 (lines 120-121): "Time continuity was also used to deal with the multiple jet issue: wind maxima associated with the same wind minimum were discarded if the double jet did not persist between two consecutive profiles."
Following that, the beginning of the sentences presenting the results (Sec. 3.1) were modified to make the connection with the methodology: (lines 184-185) "**imposing a height continuity criterion** reduced the number of detected jet…" and (line 188) "**Imposing the wind to decrease also below the jet, as in Andreas et al. (2000) or Tuononen et al. (2017)**, reduced the detected jets…".

Line 186: "… when using the same continuity criteria (Dieudonné et al., 2023)."
Dieudonné et al. use both continuity criteria (time and height) and even a direction continuity criterion. I assume that the sentence means using the same time continuity criterion of 1.5 hours and the same height continuity criterion of 50 m. Am I assuming right? If that is the case, it would be great to explicitly specify it.

Indeed, that is the case. The end of this sentence was modified accordingly (line 193-194): "when using the same detection criteria**, i.e. a 1.5-h time continuity and a 50-m height continuity**"

Lines 260-261: "… same continuity criterion …"
It's inferable from the context, but explicitly saying '… same time continuity criteria …" will make the meaning instantly clear.
The word "time" was added in this sentence.

Line 389: "… the use of different continuity criteria."
The context leads to infer that this is attributable to different threshold of the time continuity criterion. Am I assuming right?
This sentence was modified into: "the use of **different values for the time continuity criterion**"

2.  The authors are not consistent describing the minimum selection criterion.

Line 176: "Initially, a criterion on the wind decrease below the jet was also applied …".
Since wind speed always decreases below the jet peak, I infer that the authors mean "Initially, a minimum selection criterion was also applied, as seen in Figure 2." Am I inferring correctly? If that is the case, the sentence should be corrected accordingly. Additionally, the authors should consider describing the criterion here, as they did later in the caption of Figure 2.

The initial explanation was unclear; the sentence was revised to improve clarity. The criterion on the wind decrease below the jet is different from the minimum selection criterion, and consists of imposing a threshold on the absolute and relative wind speed difference between the wind maximum and the wind minimum below.

The criterion for the minimum selection is defined in section 2.2 and illustrated on Figure 2.

Lines 176-177: "… (Andreas et al., 2000; Karipot et al., 2006; Tuononen et al., 2017)."

- Andreas et al. and Tuononen et al. mirror the main criterion (that the local maximum is at least 2 m/s stronger and at least 25% stronger than the local minima above the core) but applied also to the local minima below the core. That is slightly different from what is shown in Figure 2 where the threshold is just 1 m/s. If the criterion description from both references fits better what the authors applied as 'minimum selection criterion', they should explain accordingly in the manuscript.

  We modified the sentence about this 1 m/s threshold, to make it clear that it is used to assess the wind decrease above the jet (lines 109-111): "when a minimum was followed by an increase in wind speed of less than 1 m.s$^{-1}$ before dropping again (Fig. 2), **the wind fall-off was evaluated using**  a higher-altitude minimum  instead"
  We then completed the text to better explain the link between the existence of a local minimum and the multiple jet problem (lines 112-115): "However, this could result in the false detection of multiple jets, with two consecutive wind maxima being associated with the same minimum and both fitting the fall-off criteria (Fig. 2). For this reason, other researchers (Andreas et al., 2000; Tuononen et al., 2017) imposed that the wind should also decrease below the jet core, with the same absolute and relative fall-off criteria as above the jet core. Both approaches were tested in this study."

- Karipot et al. analyzed an intermittent LLJ on "the night of June 25–26, 2004", but they don't mention the criteria used for their selection (or at least I can't find them). The authors should consider if the citation is relevant in this place.
  You are right, this citation is not relevant here and has been removed.

Line 181: "… no criterion was imposed on the wind decrease below the jet."
The assertion is not consistent with what is shown in Table 3, 3$^{rd}$ row. Maybe the authors mean that the minimum selection criterion was not used for the final conclusions. Please, confirm.

The criterion on the wind decrease below the jet was tested, but not retained for the rest of the study process. The sentence was corrected in that sense (line 191): " the criterion on the wind decrease below the jet was not retained for the rest of the study".

Table 3 (general caption and 3$^{rd}$ column header): "… and criterion on the wind decrease below the jet."; "Wind decrease below the core". Since wind speed always decreases below the jet peak, I infer that the authors mean the 'minimum selection criterion'. Am I inferring correctly? If that is the case, the authors should update accordingly.

We retained these two formulations, since this criterion was tested; the issue concerned the definition that lacked clarity.

**Minor comments:**

1. Lines 22-23: "… LLJs complicate both forecasting and energy production due to the fluctuations in wind speed and wind direction that they cause."
   The word "fluctuation" (i.e., change with time) appears imprecise here. It leads to infer that wind speeds and directions exhibit more turbulent behavior during LLJs than during unstable regimes. LLJs tend to be quite coherent structures, interrupted by intermittent bursts of instabilities. From the reference: "… during the nighttime (i.e., the hours when the LLJs are present), simulated TKE is fairly intermittent and characterized by a turbulent bursting type nature (Nunalee and Basu, 2014)." The authors should clarify if by 'fluctuation' they mean this intermittent behavior.
   In this sentence, we meant variations with height, so we changed the word "fluctuation" to "shear," which is more appropriate.

2. Lines 102-103: "… a wind speed fall-off of at least 2 ms−1 and 20 % compared to the minimum above." This is understandably a rephrasing of the the-facto criterion 'a wind speed maximum that is at least 2ms-1 and 25% faster than the next minimum above." The criterion is defined relative to the maximum strength as follows: (max-min)/max=0.2 (the 20 % criterion), which is the same as saying max=min*1.25 (the 25 % criterion). The rephrasing shouldn't give the impression that there is a fall-off from the minimum value. My personal suggestion would be, for example, "… a wind speed fall-off of at least 2 ms−1 and 20 % from the LLJ maximum to the minimum above."
   The sentence was modified according to your remark (lines 105-106): "at least 2 m.s$^{-1}$ and 20 % **from the LLJ core** to the minimum above."

3. Line 127: "ERA5 spatial resolution is 0.25°…".
   They have several products with different resolutions. Apparently, 0.25° corresponds to the high-resolution deterministic reanalysis, grided from the native resolution of 1°. That is fine, but would it be possible to clarify this?
   We added this information to the text (line 139).

4. Lines 137-139: "Three wind turbine models (Table 2) … (i) the most common offshore turbines in the North Sea, (ii) the most advanced turbine model currently available, and (iii) the turbine model projected for future uses (Global Wind Energy Council, 2024)."
   Since the information on Table 2 includes rotor diameters and hub heights, I assume that the authors are referring to the figure in page 53 of the report cited. I also assume that the case (i) corresponds to the year 2020, the case (ii) to the year 2023, the case (iii) to the year 2030, and that the hub height was calculated as the tip height minus half of the rotor diameter. May the authors confirm? If that is the case, they may consider adding the page number to the citation, to help readers to understand how numbers in Table 2 were obtained.
   You are right. The page number and years have been added to the citation.

5. Following on the previous comment, the figure in the reference shows trends worldwide, not just in the North Sea. Therefore, I suppose that the authors have additional information that the most common offshore turbines in the North Sea are the ones that were trending worldwide in 2020. May they confirm?

   We computed the dimensions of the most common wind turbines in the North Sea based on the model information from Chirosca et al. (2022) (we added the reference in the sentence and in the bibliography) and technical specifications provided by the manufacturers and available at https://www.4coffshore.com/. The mean hub height and

mean rotor diameter we retrieved were close to the values given in the GWEC report for the worldwide trend, so we chose to keep these values and added it in the sentence.

"(i) the most common offshore turbines in the North Sea **(Chirosca et al., 2022)**, (ii) the most advanced turbine model currently available **worldwide**, and (iii) the turbine model projected for future uses (Global Wind Energy Council, 2024, **p. 53, years 2023 and 2030**)."

6. Line 162: "… the shear was divided into two contributions: from the upper blade tip to the hub and from the lower blade tip to the hub."
Following on that sentence and noting in Figure 14 that positive shears are infrequent for the blade in the lower position (in both LLJ and non-LLJ situations), I assume that the term 'wind shear contribution' is the wind shear as 'felt' by an observer moving from the blade root to the blade tip. for example, for the blade in the lower position the average contributed wind shear on the blade would be cWShear = (windspeed@lowertip – windspeed@hub) / (height@lowertip – height@hub). Is my inference correct? Wind shears exert distributed loads (bending moments) that contribute mechanical shears on the blades; authors should note, however, that compounded interactions are more complex in a 3-blades wind turbine.

You are right. The wind shear is computed as follows:

$$S_U = \frac{U_{tip} - U_{hub}}{z_{tip} - z_{hub}}$$

The equation was added to the text in section 2.4.

7. Lines 189-191: "Cases presenting two simultaneous jets at different altitudes were very rare … 4.1 % of the jet profiles and 0.64 % of time." "Very rare" is maybe a too strong conclusion for a 4.1 % occurrence. That percentage, although small, is significant. More cautious phrasing would be "infrequently" or " quite uncommon".

We changed "very rare" to "quite uncommon".

8. Table 3, 4[th] column: "Jet detection height".
The caption can confuse some readers as they may infer that those are the heights where LLJs were detected. Maybe "Maximum scanning height" is more precise.

We changed this according to your remark.

9. Figure 4 caption: "The black line represents the coastline direction, with the sea located on the upper part and the land on the lower part." There are two black lines in each subfigure, one labeled 'Coastline' and the other one with no label. The labeled black line appears to be parallel to the coastline seen at the right in Figure 1b. What is the meaning of the unlabeled black line?

The coastline has a change of direction around the Dunkerque site. The caption was modified to precise this point and indicate that both black lines correspond to the local coastline. "The **two** black lines represent the **local** coastline **(that changes direction around the site)**,…"

10. Line 236: "… northeasterly and easterly LLJs followed the global distribution pattern … ".
While that appears to be true for NE LLJs (the yellow dashed line), Figure 7a apparently show a different pattern for E LLJs (the green line). The global distribution peaks on the months of June- July, while the E LLJs distribution peaks on the month of September. The description merits a reconsideration.

You are right, we modified the description to specify that the easterly jets' maximum of occurrence is in September.

11. Line 238: "… they occurred over a shorter period …"
Since the number of NW LLJs (the blue line) outside the period is not negligible, a milder claim may be more precise, something like "… they tend to occur over a shorter period …"

We changed "they occurred over a shorter period" to "they tend to occur over a shorter period".

12. Line 242: "High-speed LLJs occurred year-round, but their peak occurrence was observed in March and October …"
I infer that the authors mean WS> 15 m/s (the yellow areas). Figure 7b appears to show that those jets are mostly observed during the months of October, April, and March (in that order). Consequently, April is more prone to high-speed LLJs than March. This observation doesn't contradict the authors description that high-speed LLJs tend to peak during the periods where S LLJs are more frequent, especially considering that the frontier between April and May is artificial, but omitting April seems a little forced.
Indeed, we added April to the list.

13. Figure 8: "… which closely corresponds to the local solar time."
'closely' is imprecise. More informative is to indicate that the local time in Dunkerque is UTC+1 in normal time and UTC+2 during daylight saving time.

Actually, we meant that the solar noon is close to 12:00 UTC in Dunkerque, but we agree that the phrasing "local solar time" can be misunderstood. The legend was modified as: "solar time in Dunkerque".

14. Lines 261-262: "… with LLJs less than 3 hours apart being considered as a single event …
The longest event lasted 23.6 h, and the average duration was 5.4 h."
LLJ events can last much more than 3 hours, that is a fact. Therefore, those results are completely reasonable. The problem is the 3-hour method that is not sufficiently described. In a 23.6-hour event there are many profiles that are separated by more than 3 hours and nevertheless were considered (correctly) as part of a single event. I suppose that the method worked like a rolling window in which LLJ profiles separated by 3 hours or less were considered belonging to the same event, even if there were intermediate non-LLJ profiles. Am I assuming correctly?

The method used to discretize LLJs into events is not exactly a rolling window but rather a time discontinuity approach, where consecutive LLJ profiles separated by more than 3 hours are classified as distinct events, while smaller gaps are tolerated. We rephrased the end of the explanation sentence (lines 271-272): "wind profiles are considered to belong to the same LLJ event as long as there is no interruption longer than 3 hours".

15. Figure 10: "Distribution of the Pasquill atmospheric stability classes …"
It is interesting to see LLJs developing in extreme unstable conditions. Since the Monin-Obukhov stability parameter (used to determine the Pasquill stability classes, subsection 2.3) was calculated using measurements from the ultrasonic anemometer located at the low altitude z=15 m (line 96), maybe there is a residual layer above where LLJs develop. Or this specific LLJ type is explained by a mechanism other than Blackadar alone (for example, sea breezes).

In Dunkerque, we have an important proportion of diurnal jets (45 % of the total), which explains the large number of LLJs falling into unstable stability classes. Moreover, the

frequent occurrence of these jets during the warm season accounts for the proportion of LLJs in the moderately and extremely unstable classes.

We indeed think, as stated in the discussion, that the most frequent jet direction (NE) corresponds mostly to sea breezes – "corkscrew sea breezes" to follow the classification introduced by Miller et al. (2003) – though determining the formation mechanism with certainty from a single measurement point is challenging.

16. Lines 388-389: "… an average duration of 3.1 h, which is shorter than the 5.4 h obtained in the present study. This difference may be attributed to the use of different continuity criteria." That explanation is reasonable, as well as the explanation given soon after. On the other hand, since the longest event was also greater in the new study (23.6 h versus 20 h 50 min in Dieudonné et al.), would it be possible also that high-altitude LLJs tend to last longer?

To determine whether higher-altitude jets persist longer than lower-altitude jets, we plotted the mean core altitude distribution of jets for short (≤ 3 h) and long (> 3 h) events, based on the duration threshold used by Weide Luiz and Fiedler (2022). The figure below shows that the mean jet altitude is very similar for both short and long events (mean of 258 m and 266 m, respectively). However, the distribution for short events extends to higher altitudes, suggesting that LLJ persistence in time does not necessarily increase with height.

**Mean jet core altitude for short and long events**

[Figure]

17. Lines 406-407: "… these jets would correspond to purely local sea breeze cases, without contribution of the wind channeling into the Dover Strait." As stated soon after, there is probably a second mechanism; therefore, the phrase "would correspond to purely local sea breeze cases" to encompass all NW LLJs may result inaccurate. Since Figure 11 shows that NW LLJs occurs through a wide range of temperature gradients (both negative and positive), would it be possible that NW LLJs observed during positive temperature gradients might be due to purely diurnal sea breezes, and NW LLJs observed during negative temperature gradients might be due to purely nocturnal inertial restoration (i.e., Blackadar mechanism)?

This paragraph was rephrased, also following the remarks from reviewer #1. We attributed diurnal NW LLJs to "pure sea breezes", following the classification introduced by Miller et al. (2003), while nocturnal NW LLJs could result from frictional decoupling between the cold sea surface and the warmer air above.

18. Appendix A. Data availability.
    Figure A1 and Table A1 are positioned before the section heading.
    That was corrected.

**Bibliography:**

- Chirosca, A. M., Rusu, L., & Bleoju, A.. Study on wind farms in the North Sea area. Energy Rep., 8, 162-168, https://doi.org/10.1016/j.egyr.2022.10.244, 2022.

- Miller, S., Keim, B., Talbot, R., and Mao, H.: Sea breeze: Structure, forecasting, and impacts, Rev. Geophys., 41, https://doi.org/10.1029/2003RG000124, 2003.

- Weide Luiz, E. and Fiedler, S.: Spatiotemporal observations of nocturnal low-level jets and impacts on wind power production, Wind Energ. Sci., 7, 1575–1591, https://doi.org/10.5194/wes-7-1575-2022, 2022.

- Andreas, E. L., Claffy, K. J., and Makshtas, A. P.: Low-level atmospheric jets and inversions over the western Weddell Sea, Bound.-Layer Meteorol., 97, 459–486, https://doi.org/10.1023/a:1002793831076, 2000.

- Tuononen, M., O'Connor, E. J., Sinclair, V. A., and Vakkari, V.: Low-level jets over Utö, Finland, based on Doppler lidar observations, J. 615 Appl. Meteorol. Climatol., 56, 2577–2594, https://doi.org/10.1175/jamc-d-16-0411.1, 2017.